# The SARS-CoV-2 nucleocapsid phosphoprotein forms mutually exclusive condensates with RNA and the membrane-associated M protein

Shan Lu [1,2,7], Qiaozhen Ye[1,7], Digvijay Singh [3], Yong Cao [4], Jolene K. Diedrich[5], John R. Yates III [5], Elizabeth Villa [3], Don W. Cleveland[1,2✉] & Kevin D. Corbett [1,6✉]

The multifunctional nucleocapsid (N) protein in SARS-CoV-2 binds the ~30 kb viral RNA genome to aid its packaging into the 80–90 nm membrane-enveloped virion. The N protein is composed of N-terminal RNA-binding and C-terminal dimerization domains that are flanked by three intrinsically disordered regions. Here we demonstrate that the N protein's central disordered domain drives phase separation with RNA, and that phosphorylation of an adjacent serine/arginine rich region modulates the physical properties of the resulting condensates. In cells, N forms condensates that recruit the stress granule protein G3BP1, highlighting a potential role for N in G3BP1 sequestration and stress granule inhibition. The SARS-CoV-2 membrane (M) protein independently induces N protein phase separation, and three-component mixtures of N + M + RNA form condensates with mutually exclusive compartments containing N + M or N + RNA, including annular structures in which the M protein coats the outside of an N + RNA condensate. These findings support a model in which phase separation of the SARS-CoV-2 N protein contributes both to suppression of the G3BP1-dependent host immune response and to packaging genomic RNA during virion assembly.

[1] Department of Cellular and Molecular Medicine, University of California, San Diego, La Jolla, CA 92093, USA. [2] Ludwig Institute for Cancer Research, San Diego Branch, La Jolla, CA 92093, USA. [3] Section of Molecular Biology, Division of Biological Sciences, University of California San Diego, La Jolla, CA 92093, USA. [4] National Institute of Biological Sciences, 102206 Beijing, China. [5] The Scripps Research Institute, La Jolla, CA 92037, USA. [6] Department of Chemistry & Biochemistry, University of California San Diego, La Jolla, CA 92093, USA. [7] These authors contributed equally: Shan Lu, Qiaozhen Ye.
✉email: dcleveland@health.ucsd.edu; kcorbett@health.ucsd.edu

The ongoing COVID-19 pandemic is caused by severe acute respiratory syndrome coronavirus 2 (SARS-CoV-2), a highly contagious betacoronavirus[1,2]. Coronaviruses comprise a large family of positive-stranded RNA viruses, whose ~30 kb genome is packaged into a membrane-enveloped virion 80–90 nm in diameter[3,4]. The first two-thirds of the genome encodes two polyproteins that are processed by virally encoded proteases into nonstructural proteins, and which assemble into the viral replicase–transcriptase complex (RTC)[5]. The final third of the genome generates subgenomic RNAs encoding accessory proteins plus the four main structural proteins of the virion: the spike (S) protein that recognizes cell receptors, the nucleocapsid (N) protein responsible for viral RNA packaging, and the membrane-associated envelope (E) and membrane (M) proteins[6,7].

The RNA-binding N protein plays two major roles in the coronavirus life cycle. Its primary role is to assemble with genomic RNA into the viral RNA–protein (vRNP) complex, and mediate vRNP packaging into virions via poorly understood interactions between the N and M proteins[8–10]. Second, the N protein localizes to RTCs at early stages of infection, where it is thought to facilitate viral RNA synthesis and translation by recruiting host factors and promoting RNA template switching[11–15]. To accomplish these important functions, betacoronavirus N proteins have evolved a modular architecture with two conserved, folded domains flanked by three intrinsically disordered regions (IDRs). The N-terminal domain (NTD) is thought to mediate a specific interaction with the viral genome's packaging signal, and the C-terminal domain (CTD) forms a compact dimer that has been proposed to aid vRNP assembly[16–21]. These two domains are separated by a conserved central IDR containing a serine/arginine-rich region (SR) that is highly phosphorylated in infected cells[22–26], and are flanked by less well-conserved IDRs at the N- and C-terminus. The C-terminal IDR of the related SARS-CoV N protein has been implicated in M protein binding, suggesting an important role for this domain in viral packaging[9,10].

In isolation, betacoronavirus N proteins self-associate into dimers, tetramers, and larger oligomers that are thought to form the basis for assembly of the vRNP complex[21,27–29]. Electron microscopy analysis of several betacoronaviruses has suggested that in the virion, the N protein mediates assembly of a helical filament-like vRNP complex[3,30–32]. Recent cryo-electron tomography (cryo-ET) of SARS-CoV-2 virions has revealed a more granular vRNP structure, with each virion containing 35–40 individual vRNP complexes that each adopt a cylindrical shell-like structure ~15 nm in diameter[4,33]. Within the virion, membrane-proximal vRNPs show a characteristic orientation with respect to the membrane, suggesting that they specifically interact with the membrane-associated M protein[33]. Within the vRNP, tentative modeling based on known NTD and CTD structures suggests a specific assembly with ~800 nt of genomic RNA (30 kb ÷ ~38 vRNPs) wrapped around ~12 copies of the N protein[33]. Individual vRNPs could also form linear stacks resembling helical filaments[4,33], reconciling the apparent conflict with earlier observations and suggesting that betacoronavirus vRNPs likely adopt a broadly conserved architecture.

In recent years, many RNA-binding proteins with IDRs have been found to undergo liquid–liquid phase separation with RNA, and these biomolecular condensates are thought to orchestrate a large number of important biological processes and in some cases drive disease[34–41]. The structural features of the betacoronavirus N protein, and its diverse roles in viral RNA metabolism and virion assembly, make it tempting to hypothesize that phase separation may play a role in this protein's functions[42]. Here, we combine in vitro reconstitution approaches and cellular assays to determine the phase separation behavior of the SARS-CoV-2 N protein. We first demonstrate that RNA can induce assembly of the N protein into phase-separated condensates in vitro, and pinpoint a ~40 residue region in the central IDR with a key role in RNA-driven phase separation. Using quantitative cross-linking mass spectrometry, we found that this region and the neighboring C-terminal dimerization domain show strong intermolecular interactions in RNA-driven condensates. We also characterized the interaction between N and a soluble fragment of the M protein, which we find independently mediates N protein phase separation through an interaction with the protein's C-terminal domains. Three-component mixtures of N, M, and RNA assemble into condensates with distinct internal compartments or layers containing N + RNA or N + M, suggesting that M and RNA repel one another despite each interacting with N. This mutual exclusion often yields two-layered condensates with an internal N + RNA compartment surrounded by an outer layer of N + M. In cells, N forms condensates that incorporate RNA and the stress granule core protein G3BP1, but not other known stress granule proteins, suggesting a role for N in suppressing G3BP1-dependent innate immune responses. Overall, our data provide mechanistic insight into multiple functions of SARS-CoV-2 N driven by phase separation, including aiding viral RNA synthesis, suppressing host immune responses, and driving virion assembly through specific, but mutually exclusive interactions with genomic RNA and the viral membrane-associated M protein.

## Results

### SARS-CoV-2 N undergoes RNA-dependent phase separation.
To investigate the mechanistic basis for nucleocapsid-mediated RNA packaging, we first purified bacterially expressed recombinant full-length N protein (419 aa, 45.6 kDa). While purification in buffers with high salt concentration (1 M NaCl) yielded pure protein, purification in lower salt buffers resulted in retention of bacterial nucleic acids and aggregation of the protein as judged by size-exclusion chromatography[21]. In light of this observation and recent findings that intrinsically disordered RNA-binding proteins undergo liquid–liquid phase separation in vitro[34–41], we investigated whether the SARS-CoV-2 N protein shares this property. We generated a variant N protein with an N-terminal cysteine for specific labeling, then mixed Cy5-labeled full-length N protein with a nonspecific 17-mer ssRNA labeled with fluorescein (6-FAM). After mixing, we observed formation of spherical phase-separated condensates containing both components (Fig. 1a). Condensates formed in a specific range of RNA concentration, above which phase separation was inhibited (Fig. 1a). This so-called reentrant phase separation behavior is commonly observed in two-component systems, including RNA–protein mixtures, and is related to the ratio of binding sites in the two components in the mixture[43]. With 10 µM N protein, maximal phase separation occurred in the presence of 5 µM 17-mer RNA (85 µM nucleotide), above which phase separation was inhibited (Fig. 1a). The observation that N forms condensates with a very short RNA suggests that phase separation in this case is driven largely by multivalent protein–protein interactions. In agreement with this idea, purified N formed condensates without addition of RNA, but these were much smaller and fewer than those formed in the presence of RNA (Supplementary Fig. 1a). The N protein independently formed condensates only at very low salt concentrations (20–40 mM KCl), while addition of RNA enhanced condensate formation at salt concentrations approaching physiological (>80 mM KCl; Supplementary Fig. 1b).

We next examined the behavior of the N protein with three larger RNAs derived from the SARS-CoV-2 genome (Fig. 1b and Supplementary Table 1): (1) UTR265, the first 265 bases of the viral genome which contains the leader sequence (located at nt 20–81) that in mouse hepatitis virus was shown to bind the N

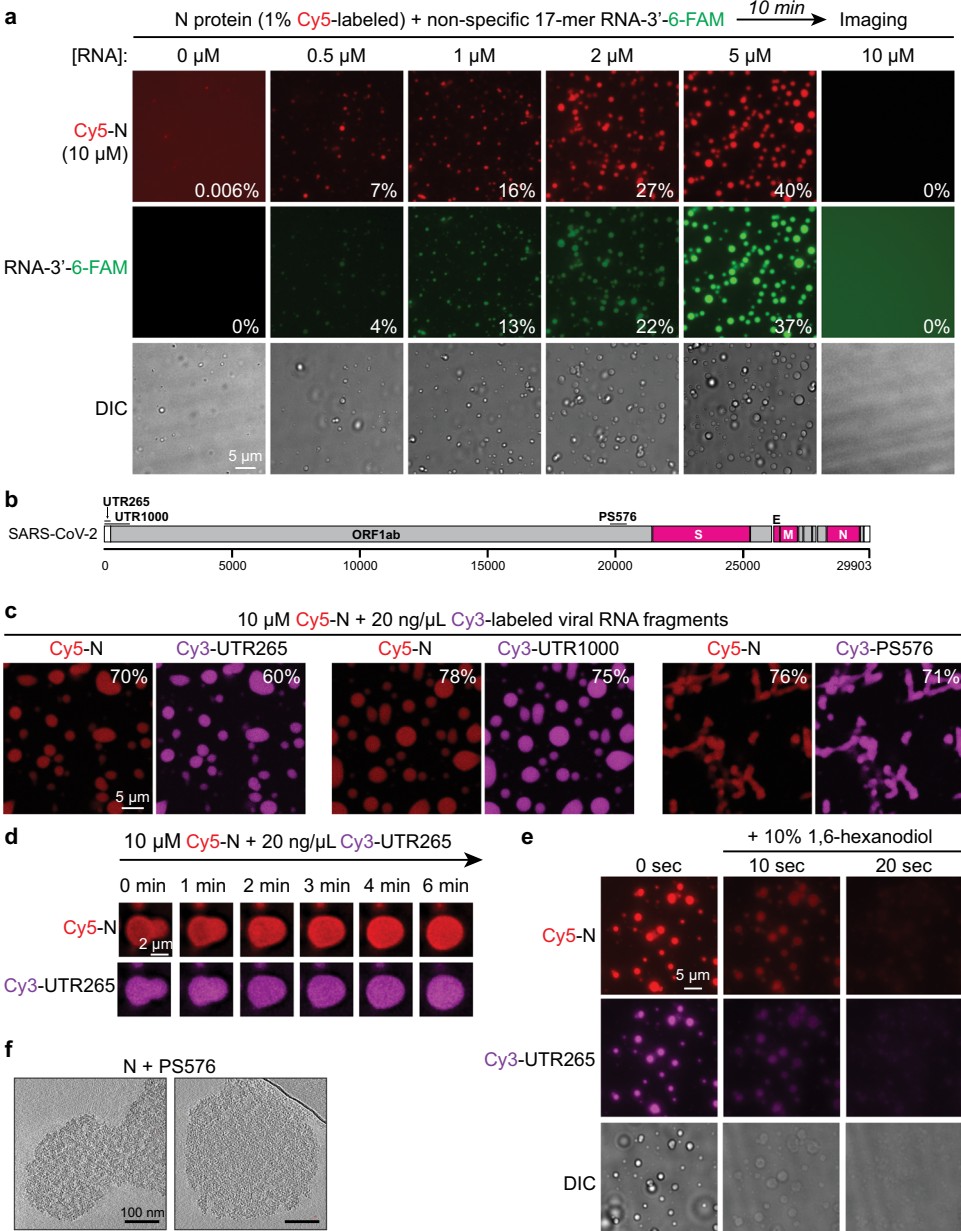

**Fig. 1 N protein undergoes phase separation with RNA. a** Fluorescence and DIC images of phase separation of N protein (1% N-terminal Cy5-labeled N protein) with a 3′ 6-FAM-labeled 17-mer ssRNA. Scale bar, 5 μm. Percentage values indicate the percentage of total RNA or protein fluorescence within condensates (see "Methods" section). See Supplementary Fig. 1a for N protein phase separation in the absence of added RNA. **b** Schematic of SARS-CoV-2 viral genome, showing the locations of four structural protein genes (pink) and three viral genome fragments used in this study (Supplementary Table 1). **c** Fluorescence images of phase-separated condensates formed by N protein with viral RNA fragments. Scale bar, 5 μm. **d** Fluorescence images of a representative fusion event of N + RNA condensates. Events were observed in 20 min after initial mixing. **e** Dissolution of N + RNA condensates by addition of 1,6-hexanediol. **f** Tomographic slice from a cryo-electron tomography reconstruction of N + RNA condensates (10 μM N protein plus 20 ng/μL PS576 RNA). Scale bar, 100 nm. See Supplementary Fig. 1f, g for power spectrum analysis of tomographic reconstructions.

protein with high affinity[44,45]; (2) UTR1000, the first 1000 bases of the viral genome that includes the 5′-UTR and part of the ORF1ab gene; and (3) PS576, a 576-nt sequence located near the end of the ORF1ab gene (nt 19786–20361) corresponding to the putative packaging signal for SARS-CoV viral RNA[46,47]. We found that addition of 20 ng/μL (~62 μM nucleotide) viral RNA fragments containing the 5′-UTR to 10 μM N protein induced the formation of round condensates (Fig. 1c). In contrast, addition of the PS576 fragment caused the formation of more amorphous structures resembling a fibrillar network (Fig. 1c). The distinct structures formed with PS576 compared to 17-mer RNA or UTR-derived RNAs suggests that the viral packaging signal may interact differently with the N protein than nonspecific RNA (see "Discussion" section). These condensates showed reentrant phase separation behavior (Supplementary Fig. 1c), grew over time (Supplementary Fig. 1d), and displayed classical behaviors, such as fusion of droplets (Fig. 1d) and dissolution upon addition of 10% 1,6-hexanediol, an organic solvent that disrupts a wide range of biomolecular condensates by reducing the hydrophobic effect[48–50] (Fig. 1e). Cryo-ET of N protein + RNA condensates revealed well-defined texture suggestive of internal order, which was also observed by super-resolution light microscopy (Fig. 1f

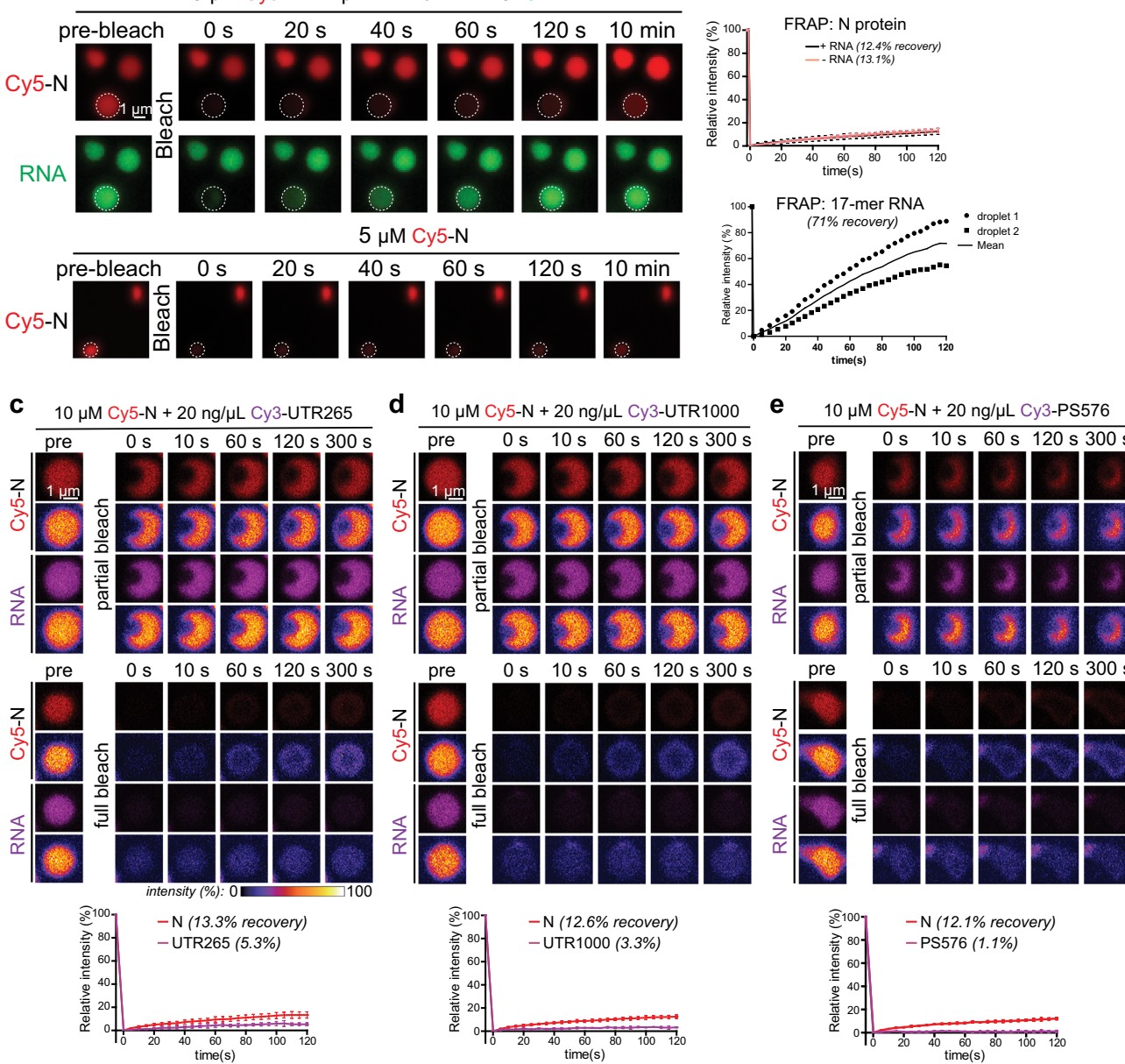

**Fig. 2 N + RNA forms condensates with slow dynamics. a, b** Representative fluorescence images (**a**) and mean fluorescence plots (**b**) of FRAP analysis of condensates formed from N protein + 17-mer ssRNA condensates (upper, $n = 4$ condensates) or N protein only (lower, $n = 2$ condensates, individual data points and average shown). Data are normalized to the average intensity of a particle before photobleaching and are represented as mean ± standard deviation from the recovery curves. Scale bar, 1 μm. **c–e** FRAP analysis of condensates formed by N protein and viral RNAs: **c** UTR265 ($n = 8$ condensates); **d** UTR1000 ($n = 8$ condensates); and **e** PS576 ($n = 3$ condensates). Fluorescence images of one partial-bleached RNA/N protein condensate and one fully bleached condensate are shown both in normal and fire coloring to enable visualization of the minimal fluorescence recovery. Mean average data are normalized to the average intensity of a particle before photobleaching and are represented as mean ± standard deviation from the recovery curves. Scale bar, 1 μm.

and Supplementary Fig. 1e–g). In particular, these condensates resemble electron-dense virus-like particles observed in the periphery of infected cells and also released alongside intact virions, which may comprise packaged RNAs that have not undergone membrane envelopment[33].

Next, we explored the biophysical properties of N + RNA condensates by fluorescence recovery after photobleaching (FRAP). Full bleaching and partial bleaching of N + RNA condensates indicated that the N protein is very slowly exchanged with the soluble pool and within the structures, with recovery of only ~12% fluorescence intensity in 2 min after bleaching (Fig. 2).

In condensates formed with the short 17-mer RNA, the RNA was highly mobile and freely exchanged (Fig. 2a, b). The mobility of this short RNA decreased with longer incubation times, indicating that initial highly dynamic condensates mature over time to a structure with slower dynamics (Supplementary Fig. 2a, b). In condensates of N protein with the larger virus-derived RNAs, both RNA and N protein showed slow dynamics (Fig. 2c, d). N protein dynamics were independent of the length of RNA in the condensates (Fig. 2c–e), but in these condensates the longer RNAs showed much slower dynamics than the N protein. These data suggest that long viral RNAs interact with multiple N

proteins and are trapped inside a framework formed by the N protein. Consistent with these slow dynamics, the fusion of N + RNA droplets was slow (Fig. 1d) and soluble N protein was incorporated very slowly into preexisting N + RNA condensates (Supplementary Fig. 2c).

**The central IDR is critical for RNA-mediated phase separation.** The N protein has multiple IDRs based on sequence alignments, disorder prediction[51], and IDR prediction algorithms including the catGRANULE server[52] (Supplementary Fig. 3a): (1) the N-terminal IDR (aa 1–48), (2) the central IDR (aa 175–246) that consists of a SR-rich region (aa 175–206) followed by a leucine/glutamine (L/Q)-rich region (aa 210–246; Supplementary Fig. 3c), and (3) the C-terminal IDR (aa 365–419). The folded RNA-binding NTD and the C-terminal dimerization domain are flanked by these three IDR regions[28,53]. In related betacoronavirus N proteins, each of these domains has been shown to bind RNA with different affinities[18,44–46,54]. To determine the contribution of different domains to the protein's ability to form phase-separated condensates with RNA, we purified 13 N protein truncations deleting the three IDRs (N-terminal, central, and C-terminal) and two folded domains (N-terminal RNA-binding domain and C-terminal dimerization domain), and tested their ability to form condensates with RNA (Fig. 3a–n, summarized in Fig. 3q). These tests revealed that several domains contribute to RNA-mediated phase separation. First, deletion of the N-terminal IDR and/or C-terminal IDR reduced, but did not eliminate, phase separation (Fig. 3b–d). Next, neither the N-terminal RNA-binding domain nor the C-terminal dimerization domain was required for phase separation, as truncations deleting either of these domains still formed condensates with RNA (Fig. 3e, g, k, l). Finally, our data reveal a critical role for the central IDR, and in particular the L/Q-rich subdomain (residues 210–246) adjacent to the S/R-rich region, in RNA-mediated phase separation. Most N protein truncations lacking the L/Q-rich region failed to form condensates with RNA (Fig. 3f, h–j). Internal deletion constructs missing only the S/R-rich region or the L/Q-rich region were able to form condensates with RNA, likely because of the collective contributions of other domains (Fig. 3m, n). Nonetheless, the striking difference in behavior between constructs that differ only in the presence of the central IDR (e.g., $N^{1-246}$ (Fig. 3e) vs. $N^{1-175}$ (Fig. 3i) and $N^{175-419}$ (Fig. 3k) vs. $N^{247-419}$ (Fig. 3f)) or the L/Q-rich region within it (e.g., $N^{49-246}$ (Fig. 3g) vs. $N^{49-209}$ (Fig. 3h)) points to a key role for the L/Q-rich region in RNA-mediated phase separation.

The S/R-rich region of the central IDR (residues 176–206) is highly positively charged with six arginine residues in 31 amino acids (Supplementary Fig. 2c). In related coronaviruses, this region becomes highly phosphorylated in infected cells[22,23,25,26], suggesting that phosphorylation could regulate the N protein's RNA-mediated condensation. To test this idea, we generated full-length recombinant N protein with all 14 serine residues in the S/R-rich region mutated to alanine ($N^{14SA}$), or 11 serine residues mutated to aspartate to mimic the negative charge associated with phosphorylation ($N^{11SD}$). Both mutants formed condensates with RNA (Fig. 3o, p). We found that droplets formed by the phosphomimetic mutant $N^{11SD}$ showed dramatically faster kinetics of droplet fusion than wild-type N or $N^{14SA}$ (Fig. 3r), suggesting that phosphorylation reduces the viscosity of N + RNA condensates. At a molecular level, this effect may arise from modulation of either RNA–protein or protein–protein interactions. This result parallels another recent report showing that mimicking S/R phosphorylation through serine-to-aspartate mutations reduces the viscosity of SARS-CoV-2 N + RNA condensates in vitro[55].

Overall, these data show that the central IDR is critical for SARS-CoV-2 N protein phase separation with RNA, and that the viscosity of the resulting condensates is likely regulated by phosphorylation in the S/R-rich region. We propose that while several domains within the N protein contribute to RNA binding and phase separation, the central IDR plays a key role in RNA-mediated condensation and regulates the physical properties of the resulting condensates.

**Quantitative cross-linking mass spectrometry reveals key interaction interfaces.** Next, we sought to identify particular interfaces or subdomains within the N protein that mediate condensation with RNA, using quantitative cross-linking mass spectrometry[56]. We used the cross-linking agent BS3 (bis(sulfo-succinimidyl)suberate), which can cross-link pairs of lysine residues within ~24 Å (Cα–Cα distance). We differentially labeled soluble N protein and RNA-mediated condensates with isotopically labeled BS3-d0 and BS3-d4, which differ in molecular weight by 4 Da due to the replacement of four hydrogen atoms with deuterium (Fig. 4a). The 4-Da molecular weight difference between BS3-d0 and BS3-d4 enables quantitative comparison of cross-linking between each pair of lysine residues in the soluble vs. condensed state (Fig. 4a and Supplementary Fig. 4a, b). We used three RNA conditions to induce N protein phase separation —40 ng/µL UTR265, 160 ng/µL UTR265, and 40 ng/µL PS576 (Supplementary Table 2)—and measured significant increases in cross-linking (more than twofold increase, $p$ value < 0.1) for 12, 30, and 15 lysine pairs, respectively, in each condition (Fig. 4b, Supplementary Fig. 4c–f, and Supplementary Data 1). In particular, we observed strong increases in cross-linking in a cluster of lysine residues spanning the L/Q-rich region of the central IDR and part of the C-terminal dimerization domain (K233–K266), and in a cluster of lysine residues in the C-terminal IDR (K369–K375; Fig. 4b, c). For most cross-linked lysine pairs, (e.g., K249–K256 or K256–K374), it is impossible to determine whether the cross-link arose from intramolecular or intermolecular interactions. We did detect two cross-links (K256–K256 and K248–K249) that were significantly enriched upon RNA-mediated condensation in multiple samples (Supplementary Fig. 4f), which because of their proximity in sequence were verifiably intermolecular cross-links. Overall, our data parallel another recent study showing strongly increased cross-linking of two regions spanning residues 235–256 and 369–390 upon condensation of the N protein alone in low salt buffer[57] (Fig. 4c), hinting that condensation with and without RNA is likely mediated, at least in part, by a similar set of protein–protein interactions.

When we mapped the residues showing strong cross-linking increases upon RNA-mediated condensation onto our recent structure of the N protein C-terminal dimerization domain, we observed that eight lysine residues (K248, K249, K256, K257, K261, K266, and to a lesser extent K342 and K347) define a large positively charged surface on the dimeric structure of this domain (Fig. 4d). We propose that this surface is likely directly involved in RNA binding, and that RNA interactions bring multiple N protein dimers together into close proximity during condensation. Finally, we note that similar cross-links are found in both low (40 ng/µL) and high (160 ng/µL) RNA concentration, even though we had earlier observed that high RNA concentrations suppress condensation (Fig. 1a and Supplementary Fig. 1c). Thus, while higher RNA concentrations suppress the formation of visible RNA–protein condensates, the mode of N protein–RNA interactions is likely similar at different RNA concentrations.

**SARS-CoV-2 N forms phosphoregulated condensates in cells.** In addition to its role in viral packaging, the N protein is

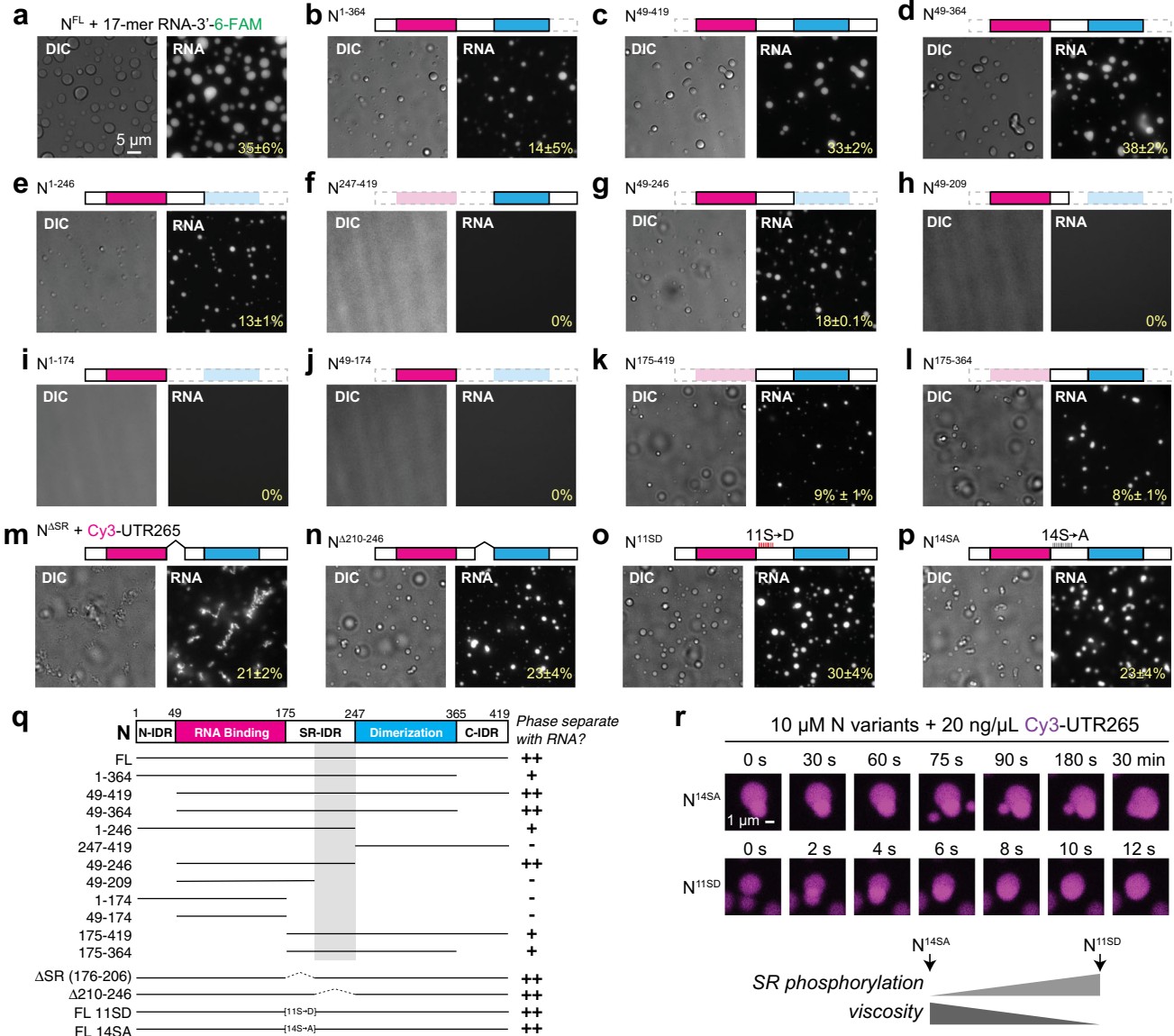

**Fig. 3 The central intrinsically disordered region is important for RNA-dependent phase separation of N protein. a–l** DIC and fluorescence images of the mixtures of 2 μM 17-mer ssRNA and 10 μM N protein variants: **a** full-length $N^{1-419}$; **b** $N^{1-364}$; **c** $N^{49-419}$; **d** $N^{49-364}$; **e** $N^{1-246}$; **f** $N^{247-419}$; **g** $N^{49-246}$; **h** $N^{49-209}$; **i** $N^{1-174}$; **j** $N^{49-174}$; **k** $N^{175-419}$; **l** $N^{175-364}$. **m–p** DIC and fluorescence images of mixtures of 20 ng/μL Cy3-UTR265 RNA and 10 μM of N variants: **m** $N^{\Delta SR}$ (note that this construct shows significant aggregation on its own; see Supplementary Fig. 3e); **n** $N^{\Delta 210-246}$; **o** $N^{11SD}$; **p** $N^{14SA}$. Images were taken 30 min after mixing N protein variants with RNA. Scale bar, 5 μm (**a–p**). Percentage values indicate the percentage of total RNA fluorescence within condensates, calculated from two (**f–m**) or three (**a–e**, **n–p**) fields. See Supplementary Fig. 3b for SDS–PAGE analysis of purified proteins, Supplementary Fig. 3d for phase separation of proteins without RNA, and Supplementary Fig. 3f for phase separation of $N^{1-364}$, $N^{1-246}$, and $N^{49-246}$ with UTR265 RNA. **q** Summary of phase separation behaviors of N protein variants shown in **a–p**. Domain schematic of the SARS-CoV-2 N protein, with known domains marked. Gray shading indicates the region implicated in N + RNA phase separation. + indicates 1–15% of RNA fluorescence within condensates; ++ indicates 16% or higher. **r** Fusion events of phospho-resistant ($N^{14SA}$) and phospho-mimetic ($N^{11SD}$) N protein droplets (representative of four fusion events observed). Imaging was performed 30 min after initial mixing. Scale bar, 1 μm.

dynamically localized to the viral RTC and is required for efficient viral RNA transcription[11–15]. Consistently, a recent proteomic study showed that SARS-CoV-2 N strongly interacts with cellular RNA processing machinery, including many stress granule proteins and several RNA helicases, including DDX1, which has been reported to be involved in viral RNA synthesis[26,58]. These data suggest that the N protein may form phase-separated condensates in cells to promote recruitment of factors that facilitate viral RNA synthesis. To assess the ability of the N protein to phase separate in cells, we generated a stably transformed human osteosarcoma U2OS cell line expressing

Clover-tagged SARS-CoV-2 N under the control of an inducible promoter (Fig. 5a). One day after induction of N protein expression, we observed N protein condensates within cells (Fig. 5a, b). FRAP analysis established that the N protein is highly dynamic in these structures, with 80% fluorescence recovery within 1 min (Fig. 5c, d). This apparent liquid-like behavior strongly contrasts to N + RNA condensates formed in vitro that show much higher viscosity, suggesting that condensate viscosity is dynamically regulated in cells.

Our in vitro analysis of $N^{11SD}$ suggested that the viscosity of N + RNA condensates is regulated by phosphorylation of the S/

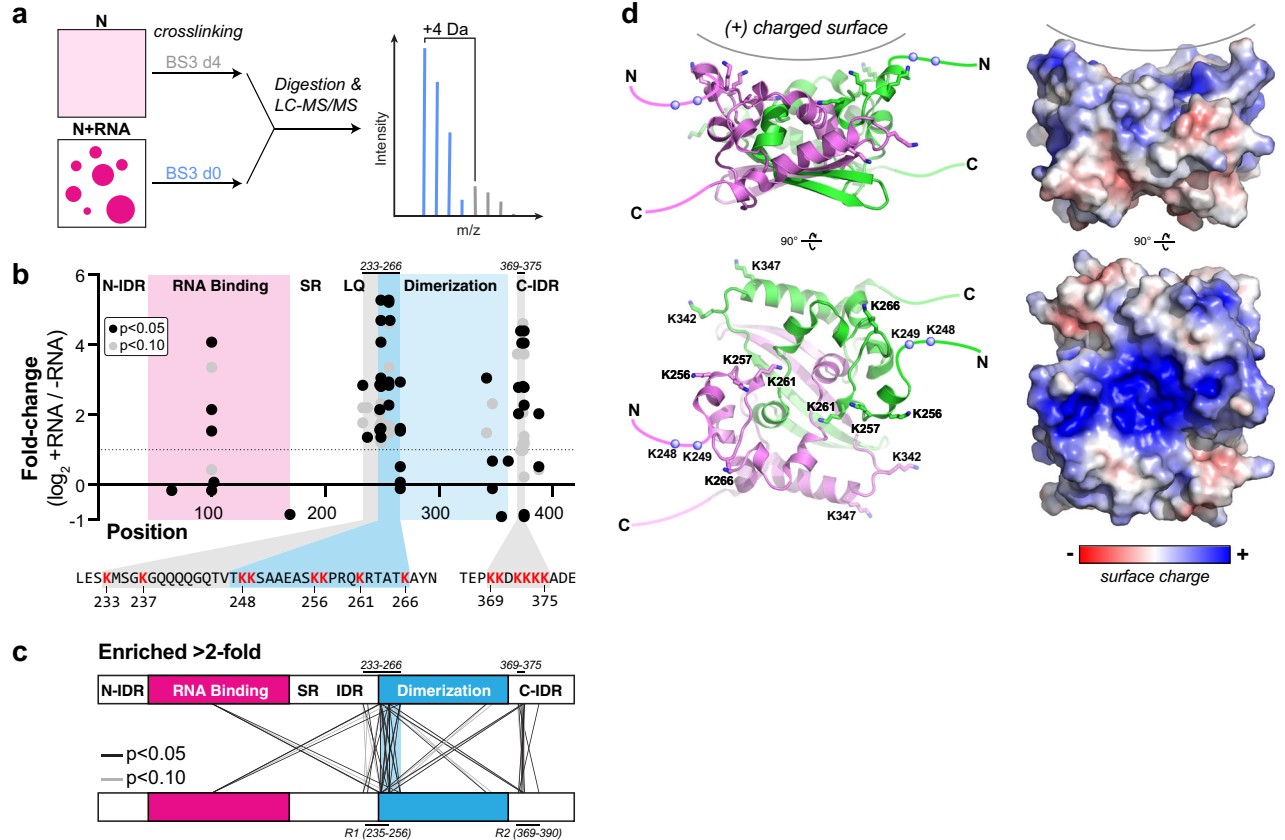

**Fig. 4 Cross-linking mass spectrometry of N + RNA condensates. a** Soluble N protein or N + RNA condensates (UTR265 or PS576 RNAs; see Supplementary Fig. 4) were cross-linked with isotopically labeled BS3, digested, and analyzed by LC-MS/MS, and each cross-linked peptide was quantitatively compared between the two conditions. **b** Graph showing the $\log_2$ fold change for each cross-linked peptide in the presence vs. absence of RNA. Average $\log_2$ fold change in three experiments is shown for each cross-linked peptide that showed a $p$ value better than 0.05 (black) or 0.1 (gray) in at least one of the three experimental condition (40 ng/μL UTR265, 40 ng/μL PS576, or 160 ng/μL UTR265 RNA). $P$ values were calculated using a one-tailed $t$ test as described in "Methods" section (see Supplementary Fig. 4 and Supplementary Data 1). Since each peptide cross-links two positions, each peptide is graphed twice. The regions spanning residues 233–266 and 369–375 (sequence shown below) show particularly strong enrichment of cross-linking. **c** Schematic showing cross-links enriched at least twofold (averaged over three experiments) that showed a confidence value better than 0.05 (black) or 0.1 (gray) in at least one experiment. Marked at top are the two regions we identified as showing strongly increased cross-linking upon RNA-mediated condensation, and marked at bottom are two regions (R1 and R2) identified by a separate study that show increased cross-linking upon N protein condensation without RNA[57]. **d** The structure of the SARS-CoV-2 N CTD (PDB ID 6WZQ[21]) reveals that residues 248–266 plus K342 and K347 form a positively charged surface on one face of the CTD dimer (K248 and K249 are disordered in the structure and are represented as blue spheres). Left: cartoon view with monomers colored green and pink; right: electrostatic surface (calculated in PyMOL using the APBS plugin[111]).

R-rich region, with phosphorylation strongly decreasing viscosity (Fig. 3r). To test whether phosphorylation of this region also regulates condensate viscosity in cells, we expressed three variant Clover-tagged N proteins in U2OS cells: the $N^{11SD}$ phospho-mimetic mutant, the phosphorylation-resistant $N^{14SA}$ mutant, and a deletion of the S/R-rich region (residues 176–206; $N^{\Delta SR}$). We found that $N^{11SD}$ behaves similarly to wild-type N in cells, forming condensates in a minority of cells (20% in wild-type N vs. 45% in $N^{11SD}$; Fig. 5b) that show fast fluorescence recovery in FRAP analysis, indicative of low viscosity (Fig. 5c, d and Supplementary Fig. 5a, b). In contrast, $N^{14SA}$ and $N^{\Delta SR}$ formed large, dense condensates in nearly 100% of cells (Fig. 5b) that show slow and incomplete fluorescence recovery in FRAP analysis (Fig. 5d and Supplementary Fig. 5c, d). We next treated cells expressing SARS-CoV N with inhibitors to SRPKs (serine–arginine protein kinases) and GSK3 (glycogen synthase kinase 3), which have both been shown to phosphorylate the N protein in related coronaviruses[24,25]. In vitro, SARS-CoV-2 N can be efficiently phosphorylated by GSK3 after priming by Cdk1 (ref. [55]). When we treated U2OS cells expressing Clover-

tagged SARS-CoV-2 N with the GSK3 inhibitor kenpaullone, or kenpaullone plus the SRPK inhibitor SRPIN340, we observed a pronounced SDS–PAGE mobility shift that was equivalent to treatment with phosphatases (Supplementary Fig. 6a, b). The marked reduction in phosphorylation of N after treatment with kenpaullone or kenpaullone plus SRPIN340 was accompanied by an increase in the fraction of cells showing N protein condensates (Supplementary Fig. 6c, d). These findings agree with our data on $N^{11SD}$ and $N^{14SA}$ mutants, and show that dephosphorylated N is more prone to condensation in cells. Together, these data strongly support the idea that the N protein is highly phosphorylated in its S/R-rich region when expressed in cells, and that this phosphorylation regulates both its propensity to form condensates and the viscosity of those condensates. The different behavior of phosphorylated vs. unphosphorylated N may reflect an adaptation to its distinct roles, with low-viscosity phosphorylated condensates promoting viral replication and host immune evasion (see below), and high-viscosity unphosphorylated condensates mediating viral RNA packaging into virions[55].

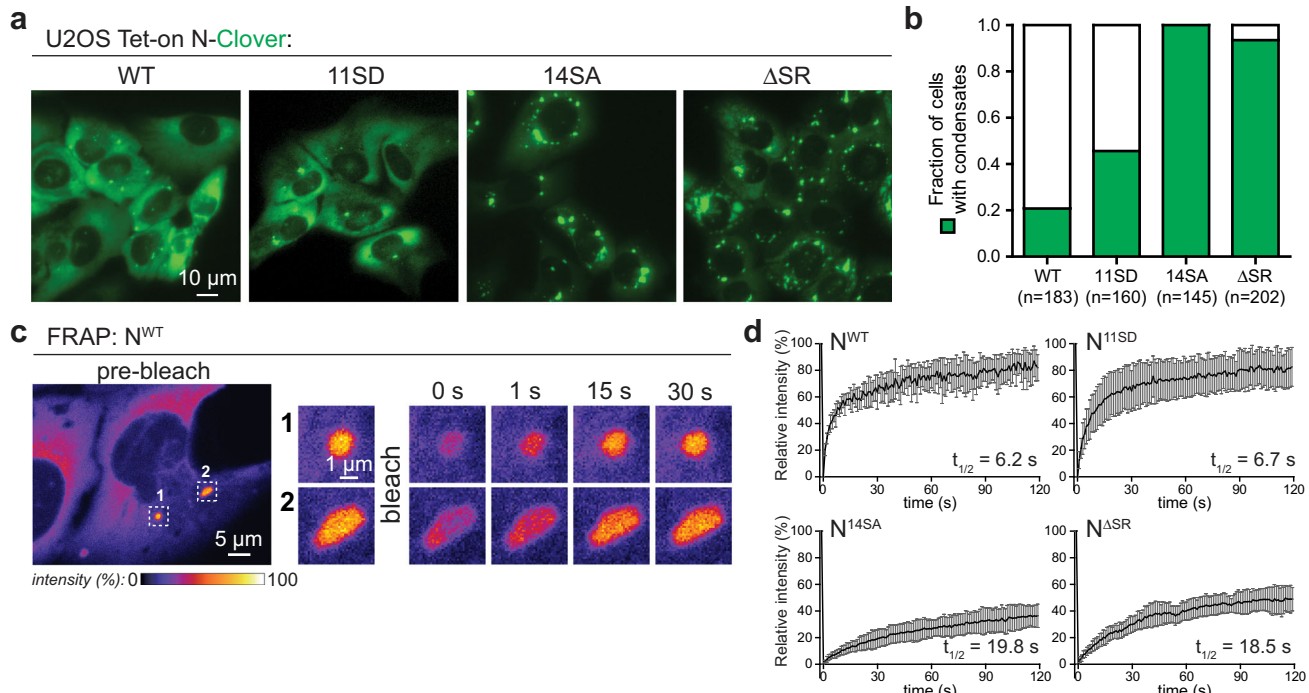

**Fig. 5 The SARS-COV-2 N protein forms highly dynamic condensates in cells whose liquidity are likely regulated by phosphorylation. a** Representative fluorescence images of Clover-tagged N, N[11SD], N[14SA], and N[ΔSR] (Δ176–206) in U2OS cells. **b** Fraction of cells showing N protein condensates when expressing Clover-tagged N, N[11SD], N[14SA], or N[ΔSR]. See Supplementary Fig. 6 for analysis with SRPK and GSK3 kinase inhibitors. N represents number of cells counted in each experiment. **c** Representative example of FRAP analysis of Clover-tagged wild-type N protein. Enlarged pictures are fluorescence images of one condensate after partial photobleaching (1) and one after full photobleaching (2). Scale bar, 5 μm for original image and 1 μm for enlarged images. **d** Mean fluorescence intensity plot for FRAP analysis of Clover-labeled N (*n* = 7 droplets), N[11SD] (*n* = 9 droplets), N[14SA] (*n* = 13 droplets), and N[ΔSR] (*n* = 10 droplets) in cells. See Supplementary Fig. 5 for full images of N[WT], N[11SD], N[14SA], and N[ΔSR]. Mean average data are normalized to the average intensity of a particle before photobleaching and are represented as mean ± standard deviation from the recovery curves.

**SARS-CoV-2 N condensates recruit the stress granule protein G3BP1.** We next sought to determine the relationship between N protein condensates and stress granules, cytoplasmic RNA:protein condensates that form in response to diverse cellular stresses[59–61]. Recently, the SARS-CoV-2 N protein has been reported to interact strongly with the stress granule core proteins G3BP1 and G3BP2 plus thousands of host mRNAs[57,58,62,63], highlighting a potential relationship between N protein condensates and stress granules. We imaged U2OS cells expressing Clover-tagged N, and observed strong colocalization with G3BP1, but not other known stress granule proteins, including UBAP2L, DDX1, and EIF3η (refs. [64,65]; Fig. 6a, b and Supplementary Fig. 7). The colocalization of N with G3BP1 is even more obvious in cells expressing phosphorylation-resistant N[14SA] and N[ΔSR] variant proteins, while the phospho-mimic mutant N[11SD] behaves similarly to wild-type protein (Fig. 6a). We next tested whether N protein condensates contain mRNA. We found that mRNAs are not enriched in the smaller condensates formed from wild-type N or the phospho-mimic N[11SD] mutant, but that mRNAs are highly enriched in the larger condensates formed from the N[14SA] and N[ΔSR] mutants (Fig. 6c). We also observed mRNA localization to wild-type N protein condensates in cells treated with the kinase inhibitors kenpaullone and SRPIN340 (Supplementary Fig. 8). Combined with our observation that condensates of N[WT] or N[11SD] are more dynamic than N[14SA] and N[ΔSR] condensates, this result suggests that mRNAs may dynamically localize to N[WT] or N[11SD] condensates, but more tightly associate with condensates of unphosphorylated N or the N[14SA] and N[ΔSR] mutants. Thus, the SARS-CoV-2 N protein forms RNA:protein condensates that recruit G3BP1, but not other stress granule proteins.

**The SARS-CoV-2 M protein independently promotes N protein phase separation.** The abundant transmembrane M protein acts as an organizational hub for virion assembly through its binding to both the membrane-anchored S protein, and to the N protein/viral RNA complex, via a soluble CTD extending into the virion[9,66] (Fig. 7a). In related coronaviruses, direct interactions between M and N have been reported, with the C-terminal IDR of N particularly implicated in this interaction[9,10]. To better understand how the SARS-CoV-2 M and N proteins interact to mediate virion assembly, we purified the soluble CTD of SARS-CoV-2 M (aa 104–222 of 222) fused to GFP (Fig. 7a and Supplementary Fig. 9a) and mixed it with purified Cy5-labeled N protein. While purified GFP-M[104–222] did not form condensates on its own (Fig. 7a) or with added RNA (Supplementary Fig. 9e), it induced formation of condensates with the N protein even in the absence of RNA (Fig. 7b). The size of these condensates increased with GFP-M[104–222] concentration and appeared as amorphous aggregates at high GFP-M[104–222] concentration, suggesting that M protein promotes condensation of N into dense gel-like assemblies (Fig. 7b and Supplementary Fig. 9b). FRAP analysis confirmed that condensates formed from GFP-M[104–222] and N were highly viscous (Supplementary Fig. 9c).

Next, we used our truncations of the SARS-CoV-2 N protein to determine the region required for phase separation with M, focusing particularly on the previously implicated C-terminal region (Fig. 7c–j, summarized in Fig. 7k). We found that removal of the C-terminal IDR reduces, but does not eliminate N + M condensate formation, while removal of the N terminal IDR (N[49–419]) does not affect condensate formation (Fig. 7c–e). Removal of the C-terminal dimerization domain and C-terminal IDR completely eliminated N + M phase separation (Fig. 7f),

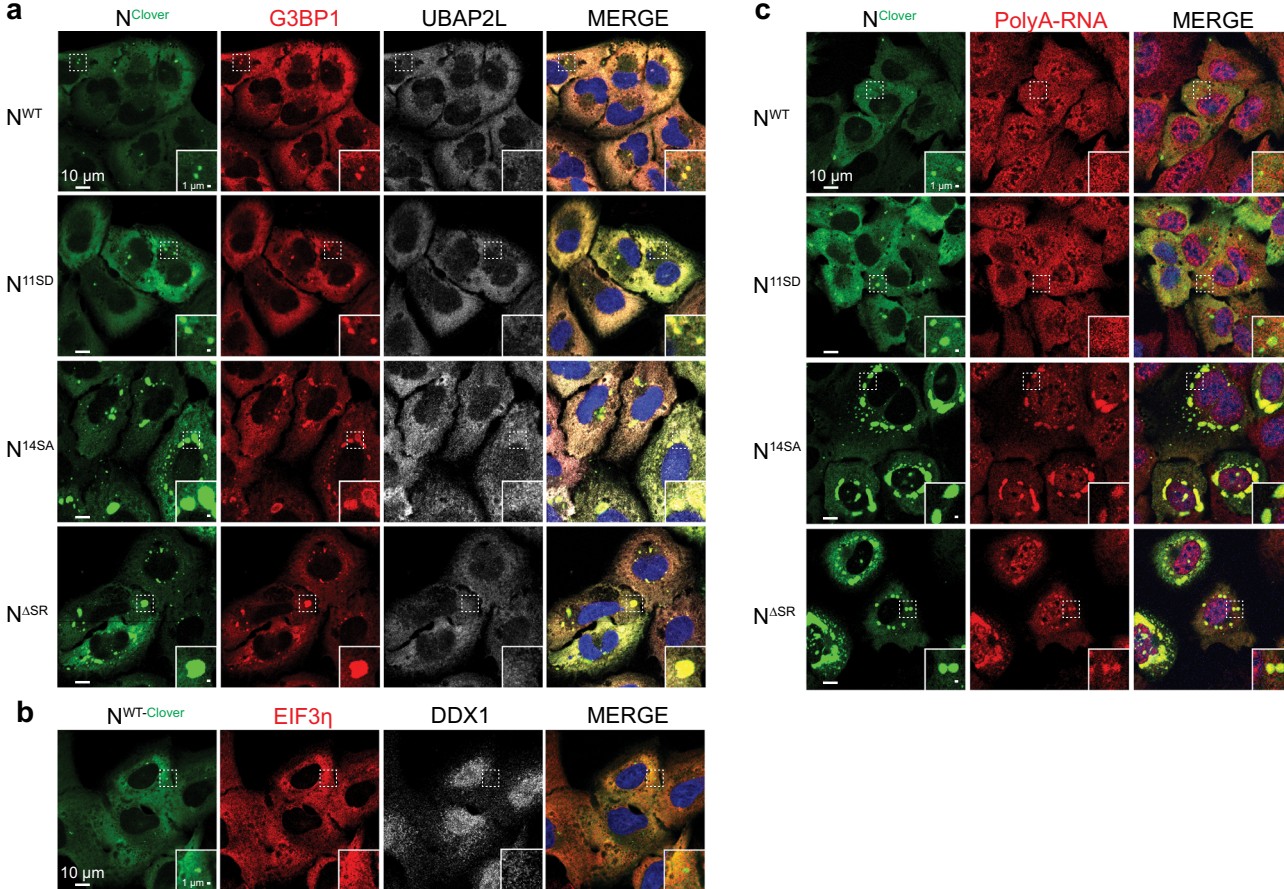

**Fig. 6 SARS-COV-2 N protein condensates recruit G3BP1, but not other stress granule proteins. a** Representative fluorescence images of Clover-tagged N protein condensates (green), plus G3BP1 (red) and UBAP2L (white). **b** Representative fluorescence images of Clover-tagged $N^{WT}$ condensates (green), plus EIF3η (red) and DDX1 (white). See Supplementary Fig. 7 for equivalent images of $N^{11SD}$, $N^{14SA}$, and $N^{\Delta SR}$. **c** Representative fluorescence images of Clover-tagged N protein condensates (green) and Poly-A RNA (oligo-dT FISH; red) from three to four images taken in each sample.

while a construct encompassing only these domains ($N^{247-419}$) was competent to form N + M condensates (Fig. 7g). An N protein fragment retaining only the C-terminal dimerization domain ($N^{247-364}$) phase separated with M, albeit to a lesser extent than $N^{247-419}$ (Fig. 7h). Addition of the central IDR to the C-terminal region did not dramatically affect N + M condensate formation (Fig. 7i, j). These results support the idea that the N protein's C-terminal region interacts with M, and that this interaction can promote the assembly of condensates in the absence of RNA. Further, RNA-mediated and M-mediated phase separation rely on distinct domains of the N protein.

Finally, we explored the behavior of three-component systems including N, M, and RNA. Within 20 min of addition of GFP-$M^{104-222}$ to preformed N + RNA condensates, GFP-$M^{104-222}$ formed an annular shell on the condensate surface that was stable for hours (Supplementary Fig. 10a). Even more strikingly, when N, M, and RNA were all mixed simultaneously, RNA and M formed mutually exclusive condensates with N. In condensates that formed with all three components, M and RNA occupied mutually exclusive subdomains, for example, with a central core of N + RNA surrounded by a shell of N + M, or vice versa (Fig. 7l and Supplementary Fig. 10b, c). FRAP analysis of these condensates showed that all three components (N protein, M protein, and RNA) show very slow dynamics (Supplementary Fig. 10d, e). Thus, the N protein forms condensates with RNA and the M protein through distinct protein regions, and N + M + RNA condensates self-organize into mutually exclusive subdomains of N + RNA and N + M.

## Discussion

A critical step in the life cycle of any virus is the packaging of its genome into new virions. This is an especially challenging problem for betacoronaviruses like SARS-CoV-2, with its large ~30 kb RNA genome. Here, we demonstrate that the SARS-CoV-2 N protein, the structural protein largely responsible for binding, compacting, and packaging the viral genome, forms phase-separated condensates with the SARS-CoV-2 M protein and RNA, including RNAs derived from the viral genome 5′-UTR and the region corresponding to the putative packaging signal of SARS-CoV. Our findings parallel recent reports of RNA-mediated phase separation by the nucleocapsid proteins from other viruses, including Measles virus and HIV-1 (refs. [67,68]). Our findings also parallel several contemporary reports of SARS-CoV-2 N + RNA phase separation[55,57,69–72]. These findings, plus recognition that many other viral nucleocapsid proteins possess domains with high predicted disorder[73], suggest a general role for phase separation in viral genome packaging and virion assembly.

Through truncation analysis, we identified a L/Q-rich region (residues 210–246) within the central IDR of SARS-CoV-2 N that plays a key role in RNA-mediated phase separation, which lies adjacent to the phosphoregulated SR-rich region (residues 176–206; Fig. 8a). A recent molecular dynamics analysis of the SARS-CoV-2 N protein's central IDR identified a putative hydrophobic α-helix spanning residues ~213–225, supporting the idea that this region may be directly involved in protein–protein interactions to promote phase separation[71]. We also found that a soluble fragment of the viral membrane-associated M protein

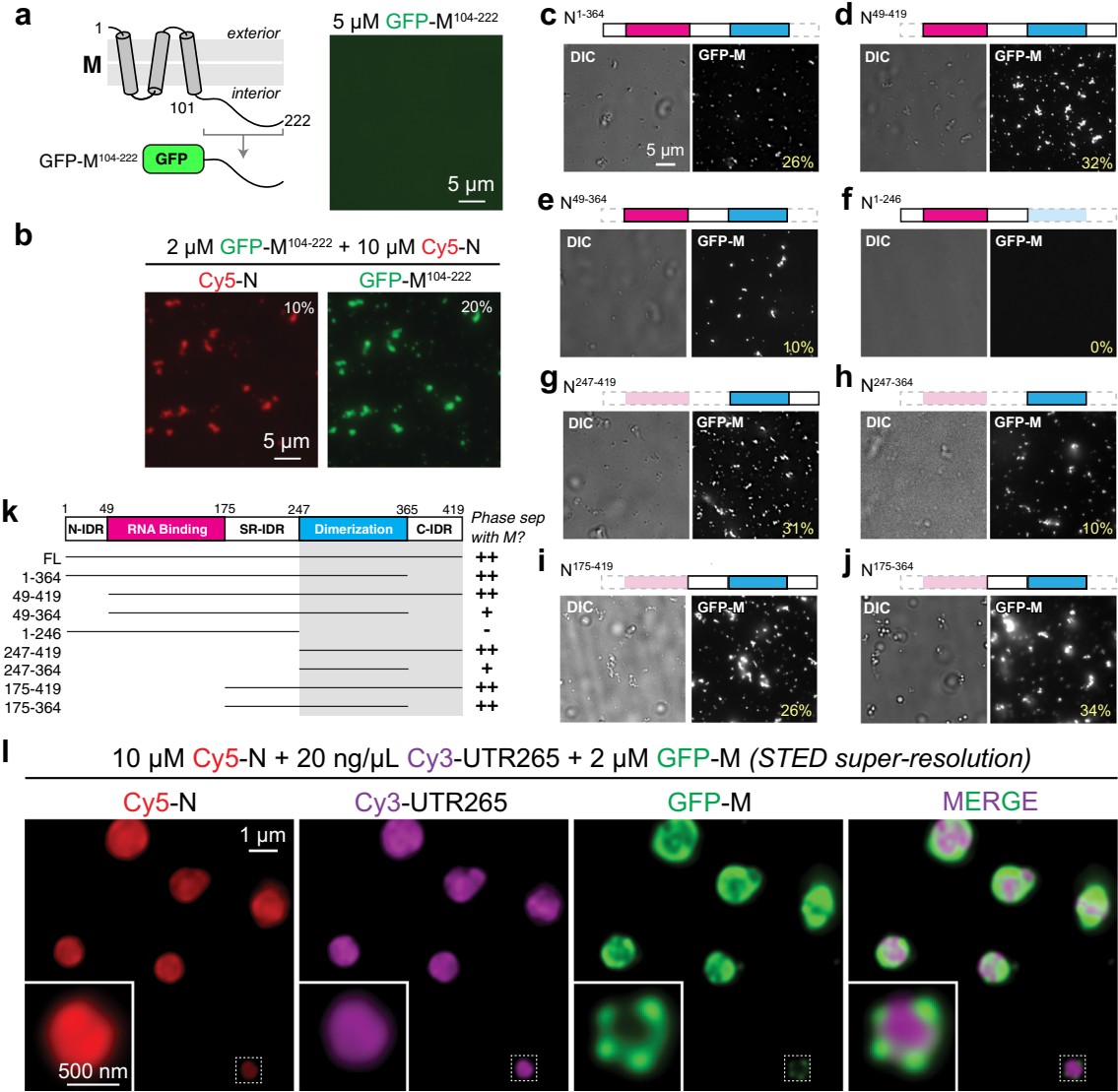

**Fig. 7 SARS-COV-2 M protein promotes N protein phase separation independent of RNA. a** Left: schematic of SARS-CoV-2 M protein showing its three transmembrane helices and C-terminal domain, and the GFP-M[104-222] fusion construct. Right: fluorescence image of 5 μM purified GFP-M[104-222]. Scale bar, 5 μm. **b** Fluorescence images of condensates formed upon mixing N (10% Cy5-labeled) with GFP-M[104-222]. Scale bar, 5 μm. GFP alone does not induce condensation of N protein (see Supplementary Fig. 9d). **c–j** Representative DIC and fluorescence images of mixtures of 2 μM GFP-M[104-222] with N protein truncations: **c** N[1-364]; **d** N[49-419]; **e** N[49-364]; **f** N[1-246]; **g** N[247-419]; **h** N[247-364]; **i** N[175-419]; **j** N[175-364]. Images were taken 20 min after mixing. Percentage values indicate the percentage of total RNA fluorescence within condensates. Scale bar, 5 μm. **k** Summary of phase separation behavior of N protein truncations when mixed with GFP-M[104-222]. + indicates 1–15% of GFP-M[104-222] fluorescence within condensates; ++ indicates 16% or higher. **l** Representative STED super-resolution images of three-component condensates formed upon mixing 10 μM N protein, 20 ng/μL UTR265, and 2 μM GFP-M[104-222]. Enlarged images show one submicron condensate with N protein/UTR265 and N protein/GFP-M[104-222] at the periphery. Scale bar, 1 μm for original image and 500 nm for enlarged figures.

interacts with N and independently induces the formation of phase-separated condensates. Three-component mixtures give rise to structures strongly reminiscent of the expected architecture of these components in virions[33,74], with a central core of N + RNA surrounded by a shell of N + M.

Our data on N–RNA and N–M interactions have important implications for understanding how the SARS-CoV-2 genome is packaged into developing virions. Recent high-resolution cryo-ET of intact SARS-CoV-2 virions has revealed that individual vRNPs adopt a characteristic shell-like architecture ~15 nm in diameter, comprising ~12 N proteins and ~800 nt of RNA[4,33]. Negative-stain electron microscopy of soluble N + RNA complexes has revealed similar particles[55], suggesting that viral RNA packaging is mediated first by assembly of individual vRNPs along the genomic

RNA, followed by condensation of these RNPs and recruitment to developing virions through interactions with the M protein at the cytoplasmic side of the ER–Golgi intermediate compartment (Fig. 8b)[4,74]. Our finding that the M protein can independently mediate phase separation of N suggests that a network of membrane-associated M proteins could mediate recruitment of N + RNA condensates into the developing virion[74].

Given the variety of RNAs in infected cells, including both host mRNAs and viral subgenomic RNAs, how is the full-length viral genome specifically recruited and packaged into virions? Current models posit that the N protein likely binds specifically to a viral RNA sequence that adopts a characteristic 3D structure. A recent computational analysis offers a compelling model for how phase separation might contribute to packaging specificity. Specifically,

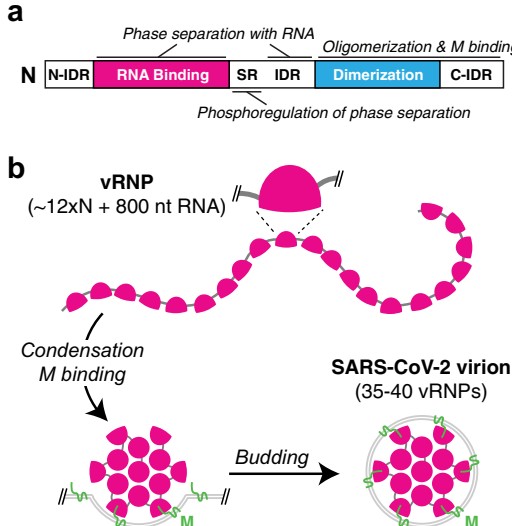

**Fig. 8 Model for SARS-CoV-2 virion assembly. a** Domain organization of SARS-CoV-2 N protein, with domains responsible for phase separation with RNA (RNA-binding domain and central IDR) and M protein (C-terminal domains) marked. Our data and related work suggests a role for phosphorylation of the SR-rich region in regulating the gel vs. liquid behavior of N + RNA condensates[55]. **b** Proposed assembly mechanism of SARS-CoV-2 virions. Viral RNPs with ~12 copies of N and 800 nt RNA coat the genomic RNA, and RNA-driven condensation plus interactions with M drive RNA packaging into virions[33]. Within the virion, individual vRNPs show characteristic orientation on the viral envelope[33] supporting a role for specific N + M interactions in packaging.

high-affinity binding of N to a particular site in the 30 kb genome could seed the assembly of vRNPs along the RNA through lower-affinity interactions, and drive condensation of stable single-genome condensates that can then be packaged into a virion[71]. This model is indirectly supported by the recent observation of dense virus-like particles ~50–80 nm in diameter in infected cells, which may represent N + RNA condensates that have not yet been enveloped[33]. The specific packaging signal is not identified for SARS-CoV-2, but may be located in the 5′-UTR or within the ORF1ab gene[44,45,69]. We find that N forms condensates with diverse RNAs, but that the morphology of condensates formed with the sequence equivalent to the putative SARS-CoV packaging signal (PS576) is distinct from those formed with other RNAs. While this observation suggests that specific RNAs could alter the phase separation behavior of N, parallel studies have shown that altering the RNA–protein ratio in condensates can give rise to similar morphological changes[55]. Further study will be required to both identify the SARS-CoV-2 packaging signal and determine how it modulates N + RNA phase separation to promote packaging of single genomes into developing virions.

In addition to their central role in viral RNA packaging, betacoronavirus N proteins participate in viral RNA transcription, especially the discontinuous transcription of subgenomic mRNAs[75]. In cells, the protein is dynamically recruited to RTCs through an interaction with viral replicase subunit NSP3 (refs. [12,76]). This activity is promoted by hyperphosphorylation of the N protein in the S/R-rich region within the central IDR, and phosphorylated N protein can recruit host factors like RNA helicases to promote viral RNA template switching and sub-genomic mRNA transcription[26]. In contrast, N protein incorporated into virions is hypophosphorylated[25,26]. We and others have shown that N protein condensates in cells show a more liquid-like behavior than those assembled in vitro, and our finding that phosphorylation of the S/R-rich region regulates

condensate viscosity is also supported by recent reports from other groups[55,69]. Given the proximity of the S/R-rich region (residues 176–206) to the L/Q-rich region we implicate in RNA-mediated phase separation (residues 210–246), it is not surprising that phosphorylation in this region could strongly affect N–protein self-interactions and alter the properties of the resulting condensates. In this manner, the physical properties of N + RNA condensates could be tuned depending on whether it is promoting viral transcription (hyperphosphorylated, low viscosity) or RNA packaging (hypophosphorylated, high viscosity).

The morphology of N protein condensates in cells is similar to that of stress granules, cytoplasmic RNA:protein condensates that form in response to stress and sequester mRNAs[34,59–61]. Stress granules are thought to play a significant role in initiating the antiviral innate immune response[77–84]. Many viruses have been shown to specifically inhibit stress granule formation, often by directly binding the stress granule core protein G3BP1, as a means to evade the host innate immune response[85–102]. We find that SARS-CoV-2 N protein condensates recruit G3BP1, but not other known stress granule proteins, including UBAP2L, DDX1, and EIF3η. These findings parallel another recent report showing that SARS-CoV-2 N sequesters G3BP1 and G3BP2, inhibits the formation of stress granules in infected cells, and alters the transcriptional program of infected cells by binding and sequestering host mRNAs[62]. In addition, we determine that host mRNA localization to N protein condensates responds to N protein phosphorylation state, with enhanced localization to condensates of unphosphorylated N protein. Thus, in addition to its roles in viral transcription and RNA packaging, SARS-CoV-2 N likely plays a role in suppressing the host innate immune response by directly targeting, and sequestering G3BP1 and host mRNAs. The importance of these roles for viral replication and immune evasion, and the regulatory role of phosphorylation in modulating the diverse activities of the N protein, are important areas for future study.

The ongoing COVID-19 pandemic demands a concerted effort to develop new therapeutic strategies to prevent new infections and lessen the severity of the infection in patients. The concurrent finding by a number of groups[55,57,69–72] that the SARS-CoV-2 N protein undergoes phase separation with RNA, and that this behavior is likely critically involved in viral RNA packaging during virion development, highlights a new step in the viral life cycle that could be targeted with therapeutics. Our findings that phase separation also mediates G3BP1 sequestration and drives interactions with the viral M protein reveals additional interactions that could be targeted to disrupt the viral life cycle. Our observations that purified N protein readily phase separates with RNA or M protein in vitro, and that N forms condensates when expressed in mammalian cells, suggest straightforward experimental strategies to test whether FDA-approved drugs or novel therapeutics may disrupt these assemblies[57]. Such a strategy could provide an important complement to current therapeutic efforts targeting other steps in the viral life cycle.

## Methods

**Cloning and protein purification**. SARS-CoV-2 N protein and truncation constructs were amplified by PCR using primers listed in Supplementary Table 3 from the IDT 2019-nCoV N positive control plasmid (IDT cat. # 10006625; NCBI RefSeq YP_009724397) and inserted by ligation-independent cloning into UC Berkeley Macrolab vector 2B-T (AmpR, N-terminal His₆-fusion; Addgene #29666) for expression in *E. coli*. N[11SD], N[14SA], N[ΔSR] (176–206), and N[Δ210–246] were generated using Gibson assembly. In N[11SD], we mutated S176, S180, S183, S184, S186, S187, S188, S190, S193, S194, S197 to aspartate and in N[14SA], we mutated S176, S180, S183, S184, S186, S187, S188, S190, S193, S194, S197, S201, S202, S206 to alanine. The gene expressing M protein residues 104–222 was synthesized by IDT, and inserted into UC Berkeley Macrolab vector 1GFP (KanR, N-terminal His₆-GFP fusion; Addgene #29663). Plasmids were transformed into *Escherichia coli* strain Rosetta 2(DE3) pLysS (Novagen) except for GFP-M[104–222] (expressed in

LOBSTR (Kerafast)) and $N^{1-364}/N^{175-364}/N^{175-419}$ (expressed in SOLUBL21 (Genlantis)), and grown in the presence of ampicillin or kanamycin and chloramphenical to an $OD_{600}$ of 0.8 at 37 °C, induced with 0.25 mM IPTG, then grown for a further 16 h at 18 °C prior to harvesting by centrifugation. Harvested cells were resuspended in buffer A (25 mM Tris-HCl pH 7.5, 5 mM $MgCl_2$ 10% glycerol, 5 mM β-mercaptoethanol, and 1 mM $NaN_3$) plus 1 M NaCl ($N^{FL}$, $N^{1-364}$, $N^{49-419}$, $N^{49-364}$, $N^{11SD}$, $N^{14SA}$, $N^{\Delta SR}$ (176–206), and $N^{\Delta 210-246}$) or 500 mM NaCl (all other constructs) and 5 mM imidazole pH 8.0. For purification, cells were lysed by sonication, then clarified lysates were loaded onto a $Ni^{2+}$ affinity column (Ni-NTA Superflow; Qiagen), washed in buffer A plus 300 mM NaCl and 20 mM imidazole pH 8.0, and eluted in buffer A plus 300 mM NaCl and 400 mM imidazole. For cleavage of $His_6$-tags, proteins were buffer exchanged in centrifugal concentrators (Amicon Ultra, EMD Millipore) to buffer A plus 300 mM NaCl and 20 mM imidazole, then incubated 16 h at 4 °C with TEV protease. Cleavage reactions were passed through a $Ni^{2+}$ affinity column again to remove uncleaved protein, cleaved $His_6$-tags, and $His_6$-tagged TEV protease. Proteins were concentrated in centrifugal concentrators and purified by size-exclusion chromatography (Superose 6 Increase 10/300 GL or Superdex 200 Increase 10/300 GL; Cytiva) in gel filtration buffer (25 mM Tris-HCl pH 7.5, 300 mM NaCl, 5 mM $MgCl_2$, 10% glycerol, and 1 mM DTT). Purified proteins were concentrated and stored at 4 °C for analysis.

For fluorescent labeling, we inserted a cysteine residue between the N-terminal $His_6$-tag and the N-terminal residue of the full-length N protein, and purified the protein as above. We labeled with either Alexa Fluor 488 (Thermo Fisher) or sulfocyanine 5 (Lumiprobe) according to Lumiprobe's protocol "Maleimide labeling of proteins and other thiolated biomolecules". Briefly, an excess of TCEP (tris-carboxyethylphosphine, up to 100× molar) was added to protein between 1 and 10 mg/mL, and kept the mixture for 20 min at room temperature, then added 1/20× fold dye solution, mixed well and left overnight at 4 °C. The next day, proteins were separated from unreacted fluorophores by passing over a Superose 6 Increase 10/300 GL column (Cytiva).

For identification of proteolytic cleavage sites, purified $N^{1-246}$ was separated from its proteolytic cleavage product by SDS–PAGE, and the band representing the cleaved product was extracted from the gel. The protein was subjected to complete trypsinization and analyzed on an Orbitrap Fusion mass spectrometer (Thermo Fisher Scientific) to identify peptides.

**In vitro transcription**. DNA templates for viral RNA fragments were synthesized (IDT) and amplified by PCR through primers with the T7 promoter sequence (TAATACGACTCACTATAGGGAG) in the 5′ end. Genomic RNA fragments (Supplementary Table 1) were synthesized using a HiScribe T7 high yield RNA synthesis kit (NEB #E2040S) with 1 µg purified PCR products. To label RNA, 0.1 µL of 10 mM Cy3-UTP (Enzo Life Sciences #42506) was added to the transcription system. After DNase I treatment, RNAs were purified using Trizol/chloroform isolation method and solubilized with 5 mM HEPES, pH 7.5 RNase-free buffer. The size and purity of RNAs were verified by denaturing urea/PAGE, and their concentrations measured by absorbance at 260 nm.

**In vitro phase separation assays**. In vitro phase separation assays were performed at 20 °C unless otherwise indicated. Unlabeled N protein was mixed with Cy5-labeled or Alexa-488 labeled N protein at 1:100 or 1:10 ratio, then used for phase separation assays. Phase separation of N protein (in 5 mM HEPES, pH 7.5, 80 mM KCl) was induced by either adding RNA (in 5 mM HEPES, pH 7.5) or M protein (in 5 mM HEPES, pH 7.5, 80 mM KCl). Samples were mixed in protein LoBind tubes (Eppendorf, 022431064) and then immediately transferred onto 18-well glass-bottom chamber slides (iBidi, 81817). Condensates were imaged within 10–20 mins or otherwise as indicated in the experiment.

**Turbidity assay**. Samples were mixed in protein LoBind tubes as in the previous section. After 30 min, the turbidity of each sample was measured by absorbance at 350 nm.

**Microscopy of in vitro phase-separated condensates**. DIC and fluorescence images of in vitro condensates were taken on a DeltaVision Elite microscope (Cytiva) with a 60× oil N.A. 1.42 objective. The laser power and exposure time of each channel are optimized to avoid saturation. Alternatively, condensates were imaged using Zeiss LSM880 or Leica SP8 STED super-resolution confocal microscopes. Laser power for imaging, digital offset and gain values are optimized for each channel to ensure the intensity lies in the linear range. For STED super-resolution imaging, image resolution settings are optimized to reach ~20 nm/pixels; average of the intensity from 16 images to increase the signal to noise ratio; 50% of 775 nm STED depletion laser is used for STED imaging. Time lapse imaging was used to capture fusion of RNA/N protein condensates on a DeltaVision microscope at 1 min intervals. For quantitation (Figs. 1, 3, and 5), condensates were identified by threshold analysis in ImageJ, then the total fluorescence inside condensates was calculated and divided by the total fluorescence in the field to yield a percentage value.

**1,6-Hexanediol treatment of phase-separated condensates**. After mixing Cy5-labeled N protein and Cy3-labeled UTR265 RNA to generate condensates, 5 µL

samples were deposited on one well of a channel slide (iBidi, 80606). A total of 20 µL of buffer containing 10% hexanediol was then added to the opposite well, and condensates were imaged at 10 s intervals on a DeltaVision Elite microscope (Cytiva) with a 60× oil N.A. 1.42 objective lens.

**Stable human cell line construction**. For expression in U2OS cells, SARS-CoV-2 N proteins (wild-type, 11 SD, 14SA, or ΔSR (176–206)) were cloned into a third-generation Tet-on system on a lentivirus vector with a Clover tag at the C-terminal of N protein. Lentivirus is produced in HEK293t cells by transfection of lentiviral plasmid and packaging plasmids pMD2.G and psPAX2. After 2 days of transfection, the culture medium containing the lentivirus was passed through a 0.45 µm filter and was used to infect U2OS cell line. After 2 days of infection, the medium is exchanged to medium containing 20 µg/µL blasticidin for selection.

**Immunofluorescence**. For immunofluorescence, U2OS cells were cultured on eight-well chamber slides (iBidi, 80827) in DMEM supplemented with 10% fetal bovine serum (FBS) and Antibiotic–Antimycotic (Thermofisher, 15240062). After treatments of the cells as indicated, cells were fixed with 4% PFA in PBS and permeabilized with 0.2% Triton X-100 for 10 min. After blocking with 1% BSA in PBS, 0.05% Triton X-100 for 2 h, cells were incubated for 1 h at room temperature with primary antibody in blocking solution. After three washes with PBS, cells were incubated with Alexa647-labeled or Cy3-labeled secondary antibody at 1:500 dilution in blocking solution for 30 min at room temperature. After three washes with PBS and DAPI staining, cells were kept in PBS for imaging. Dilutions for primary antibodies used in this study are 1:500 for anti-G3BP1 (mouse monoclonal, Abcam, ab56574), 1: 500 for anti-UBAP2L (rabbit polyclonal, Bethyl, A300-533A), 1: 100 for anti-EIF3η (goat polyclonal, Santa Cruz, sc-16377), and 1: 250 for anti-DDX1 (mouse monoclonal, Novus, NBP2-61745).

**RNA fluorescence in situ hybridization**. All hybridization steps were performed under RNase-free conditions following the Stellaris RNA fluorescence in situ hybridization (FISH) protocol for adherent cells. Briefly, cells were fixed in 3.7% formaldehyde for 10 min and permeabilized with ethanol 70% for 8 h at 4 °C. Then, cells were washed with wash buffer A (Biosearch Technologies, SMF-WA1-60) supplemented with 10% deionized formamide (Sigma, F7503), and incubated in the dark at 37 °C for 4 h in a humidified chamber with hybridization buffer (SMF-HB1-10) supplemented with 10% deionized formamide and 1 ng/ml Cy5-labeled Cy5-(d)T20 oligonucleotides (gift from Dr. J. Paul Taylor, St. Jude Children Hospital). After cells were washed with wash buffer A in the dark at 37 °C for 30 min, cells were stained with DAPI in wash buffer A, and then washed once with wash buffer B (Biosearch Technologies, SMF-WB1-20) before imaging.

**Phos-tag SDS–PAGE and western blot**. U2OS cells were cultured on 12-well plates as above. Cells were lysed in RIPA buffer supplemented with 1× protease inhibitor and phosphatase inhibitor (Thermo fisher, 78442), except for the sample to be dephosphorylated, cells were lysed in RIPA buffer supplemented with 1× EDTA-free protease inhibitor (Roche, 04693159001). Protein concentration was measured by BCA analysis (Thermo Fisher, 23227). Dephosphorylation was performed by adding 30 units of Calf Intestinal Phosphatase (New England Biolabs, M0290) to 20 µg of protein lysate supplemented with 10 mM $MgCl_2$, and incubating at 25 °C for 20 min. After adding LDS protein loading buffer and 50 mM DTT, samples were loaded onto a 7% SDS–polyacrylamide gel containing 50 µM $Mn^{2+}$-Phos-tag acrylamide (Fujifilm Wako Chemicals), and run at 180 volts for 90 min until the 70 kD marker reached the bottom of the resolving gel. After electrophoresis, the gel was immersed in transfer buffer containing 5 mM EDTA two times for 10 min each, followed by transfer buffer without EDTA. The semidry method was used to transfer the proteins to PVDF membrane. Then Clover-tagged N protein were detected by anti-GFP antibody (1:2000 dilution in 3% BSA TBST, Clontech, 632381).

**Fluorescence recovery after photobleaching**. FRAP analysis of condensates in vitro was performed on a Zeiss LSM880 Aryscan microscope with 63×/1.42 oil objective or 40×/1.2 W objective or on Nikon Eclipse Ti2 A1 confocal microscope as indicated. The intensity of the fluorescent signal is controlled in the detection range through changing the laser power, digital gain, and offset. For green, red, and far-red fluorescent channels, bleaching was conducted by 488, 561, or 633-nm line correspondingly, and the laser power and iteration of bleaching were optimized to get an efficient bleaching effect. Fluorescence recovery was monitored at 2 or 4 s intervals for 4 min. In the focal-bleach experiment, roughly half (partial bleach) or all (full bleach) of a condensate was photobleached to determine the molecular mobility with diffuse pool or inside a condensate.

U2OS cells for FRAP experiments were cultured on a eight-well chamber slide (iBidi, 80827) in DMEM supplemented with 10% FBS and Antibiotic–Antimycotic (Thermofisher, 15240062). $N^{Clover}$ expression was induced for 24 h by adding 1 µg/mL doxycycline to the culture medium. FRAP experiment was conducted as for in vitro condensate analysis using a Zeiss LSM880 microscope.

The FRAP data were quantified using ImageJ or Zeiss Zen built-in profile model. The time series of the fluorescence intensity of condensates were calculated. The intensity of the condensate during the whole experiment was normalized to

the one before bleaching and the intensity of the granule just after bleaching was normalized to zero. At least 2–10 condensates per condition were analyzed to calculate the mean and standard deviation. The averaged relative intensity and standard error were plotted to calculate dynamics.

**Quantitative cross-linking mass spectrometry**. Before cross-linking, 10 μM N protein (in 5 mM HEPES, pH 7.5, 80 mM KCl) was mixed with buffer (in 5 mM HEPES, pH 7.5) or with 40 ng/μL UTR265, 40 ng/μL PS576, or 160 ng/μL UTR265 RNAs. After incubating 10 min at room temperature, 0.5 mM BS3-d0 or BS3-d4 from a 10 mM stock was mixed with samples from each condition, as indicated in Supplementary Table 2 and incubated at room temperature for 1 h. After that, 20 mM NH$_4$HCO$_3$ was added to quench the reaction. Equal amounts of samples with and without RNA and reacted with BS3-d0 vs. BS3-d4 were mixed together before acetone precipitation and redissolving in 8 M urea, 0.1 M TEAB, pH 8.0. After reduction with 5 mM TCEP and alkylation with 10 mM iodoacetamide, three volumes of 0.1 M TEAB was added to each sample before adding trypsin at 1:50 enzyme to substrate ratio.

Digested peptides were injected onto a 25 cm, 100 μm ID column packed with BEH 1.7 μm C18 resin. Samples were separated at a flow rate of 300 nL/min on an μHPLC Easy nLC 1200 with a 120-min gradient of buffer A (0.1% formic acid in water) and buffer B (0.1% formic acid in 90% acetonitrile). Specifically, a gradient of 1–25% B over 100 min, an increase to 40% B over 20 min, an increase to 100% B over another 10 min, and a hold at 100% B for a final 10 min was used for a total run time of 140 min. The column was re-equilibrated with 20 μL of buffer A prior to the injection of each sample. MS1 and MS2 spectra were collected on an Orbitrap Eclipse using a data-dependent mode. Briefly, parameters are set as follows: for MS1 scan, 120 K resolution, 375–1800 *m/z* scan range, 1 × 10$^6$ AGC target; for MS2 scan, 60 K resolution, 0.7 *m/z* isolation window, HCD fragmentation at 35% collision energy with a 50 s dynamic exclusion window and charge 1$^+$ ions rejection.

Cross-linked peptides (Supplementary Data 1) were identified using pLink software[56,103] searching against N protein sequence with BS3-d0 and BS3-d4 set as cross-linkers, trypsin as the digestion enzyme and cysteine carboxymethylation as fixed modification. For each sample, spectra of BS3-d0 and BS3-d4 cross-linked peptides were used for quantification by pQuant software[104]. The results of three biological replicates for each condition were combined and *p* values calculated using a one-tailed *t* test by comparing the mean of the log$_2$ ratio in three biological replicates of each group to 0. Cross-link data visualization at protein bar mode was performed with xiNET[105].

**Cryo-electron tomography**. For cryo-ET, phase separation was induced by mixing 10 μM N protein (in 5 mM HEPES, pH 7.5, 80 mM KCl) with 20 ng/μL of the indicated RNA (in 5 mM HEPES, pH 7.5) for 0.5–1 min at 20 °C. A total of 4 μL of the resulting solution was deposited on glow-discharged Quantifoil grids (R2/1, Cu 200-mesh grid, Electron Microscopy Sciences). The solution on the grid was then blotted using a Vitrobot (Thermo Fisher Scientific) with conditions set to blot force −10, blot time 2.5–3.5 s, and drain time 2 s. After blotting, the grid was immediately plunge-frozen into liquid ethane/propane mixture (Airgas) cooled to close to liquid nitrogen temperature. Cryo-ET imaging was performed on a Titan Krios operated at 300 KeV (Thermo Fisher Scientific) equipped with a post-column Quantum energy filter (Gatan). The images were recorded on a K2 Summit direct detector (Gatan) in counting mode using SerialEM[106]. The tilt series were acquired using a dose-symmetric scheme[107] with a tilt range of ±60° in EFTEM mode with a nominal magnification of 42,000×–64,000× (pixel sizes at the camera 0.34–0.22 nm), tilt increment: 2–3°, and a target defocus of −3–5 μm. Motion-correction and dose-weighting, using MotionCor2[108], was applied to the individual frames of the tilt-series images. The motion-corrected and dose-weighted tilt-series images were then aligned in IMOD[109] using patch-tracking. 3D CTF correction was performed using novaCTF[110]. The weighted back-projection method was used for the final tomographic reconstruction. Power spectrum analysis of tomographic slices or individual images from tilt series was performed with IMOD.

**Statistics and reproducibility**. Images in Fig. 1a, c–e were done in one experiment. Figure 1f is representative of five tomographs. Images in Fig. 3f–h are representative of at least two images from each sample (one experiment). Events in Fig. 3r is representative of at least three events observed. Images in Fig. 6b, c represent three to four images taken in each sample (one experiment). Images in Fig. 7a–j were done in one experiment. Images in Fig. 7i represent three to four images taken in this sample. Supplementary Fig. 1d was done in one experiment and representative of two images is shown for each condition. Supplementary Fig. 1e represents one of three images taken in this condition. Supplementary Fig. 1f, g is representative of five tomographs. Supplementary Fig. 2c was done in one experiment. Supplementary Fig. 3f represents one of the two images taken for each condition. Supplementary Fig. 6b was done in one experiment. Supplementary Fig. 6c was done in two independent experiments. Images in Supplementary Figs. 7a–c and 8a–d represent three to four images taken in each sample. Supplementary Fig. 9a, b were done in one experiment. Supplementary Fig. 9d represents one of the three images taken in this sample. Supplementary Fig. 9e represents one of the two images taken in

each sample. Supplementary Fig. 10a, b represent one of the two images taken in this sample. Supplementary Fig. 10c represents one of the seven droplets in this experiment.

**Reporting summary**. Further information on research design is available in the Nature Research Reporting Summary linked to this article.

## Data availability
Source data are provided with this manuscript. Other data are available from the corresponding authors upon reasonable request. Source data are provided with this paper.

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

## Acknowledgements

We thank Jennifer Santini at the UCSD Microscopy Core and Eric Griffis at the UCSD Nikon Imaging Center for assistance with imaging, and Huilin Zhou for assistance with mass spectrometry. We are grateful for helpful discussions and suggestions from Dr. Meng-Qiu Dong, Michael Baughn, Alexander Goginashvili, Prasad Trivedi, Haiyang Yu, Sonia Vazquez-Sanchez, and Pablo Lara Gonzalez. D.S. is supported by the Damon Runyon Cancer Research Foundation (DRG-#2364-19). E.V. acknowledges funding from the NIH (DP2 GM123494), K.D.C. acknowledges funding from the NIH (R01 GM128464), D.W.C. acknowledges support from the NIH (R01 NS27036) and the Nomis Foundation, and J.R.Y. acknowledges support from NIH (P41 GM103533). We acknowledge the UCSD School of Medicine Microscopy Core Grant P30 NS047101. We acknowledge instrumentation support for mass spectrometry from NIH S10 OD023498. We acknowledge the use of the UC San Diego cryo-EM facility, which was built and equipped with funds from UC San Diego and an initial gift from Agouron Institute.

## Author contributions

S.L. conceived of the project and planned the experiments with advice from K.D.C. and D.W.C. S.L. performed all microscopy experiments, Q.Y. purified all proteins, D.S. performed cryo-electron tomography experiments, J.K.D. ran cross-linking mass spectrometry samples, and Y.C. assisted with quantitative cross-linking data analysis. All authors interpreted data. S.L. and K.D.C prepared figures, and wrote the manuscript with input from Q.Y., D.S., C.Y., E.V., and D.W.C. E.V., J.R.Y., K.D.C., and D.W.C. obtained funding.

## Competing interests

The authors declare no competing interests.
