## [Peer Review File · Nature Communications]

REVIEWER COMMENTS

Reviewer #1 (Remarks to the Author):

The authors show that N protein association and phase separation is dependent on both RNA and KCl concentration. RNA oligomers and long 5' UTR containing RNAs lead to spherical complexes. RNA with CoV packing signal lead to very large fibrillar structures. RNA oligomers can exchange from the N phase but N protein diffusion is restricted. The authors then show that the soluble domain of SARS CoV2 M protein also induces formation of a novel phase separation. The authors present striking micrographs of multi-layer particles with an external layer enriched in M and an internal layer enriched in RNA. With a series of truncation mutants the authors demonstrate that the phase separation activity can be attributed to the intrinsically disordered region (IDR) between the N-terminal and C-terminal folded domains. The authors propose a model of assembly where M helps organize N for binding to RNA and together they form the core of the nascent virus. Conversely, when expressed in mammalian cells, N protein does not form the sort of phase separation observed in vitro. They suggest that the intracellular N may be phosphorylated in the SR-rich segment of the IDP that is involved in association and may thus prevent the phase separation seen in the unphosphorylated protein in vitro.

Several groups have investigated the ability of SARS-COV2 N protein to undergo phase separation. The results demonstrated in this paper are timely and of interest. The experiments and results are intriguing. While I have many questions, I am enthusiastic about the manuscript.

While short oligos unsurprisingly diffuse in and out of the condensed phase, it seems unlikely that the larger RNAs can, but that should be ascertained. While short RNA oligos and longer RNAs from the 5'UTR yield round complexes, PS576, a 576 nucleotide RNA containing a putative packing site, yielded huge fibrillar structures. This effect is not investigated further. Will a larger RNA with the packaging site yield round complexes? Is the PS576 RNA more folded and compact? Are the PS576 complexes more less labile, or conversely can PS576 displace other RNA? It is not clear why the authors did not further investigate this RNA. Indeed, if an RNA with a packaging site yields a fibrillar structure, are the spherical particles artifactual? Will a combination of PS576 and M yield a fibrillar or compact particles.

In cells, N forms highly dynamic condensates. While under selected experimental conditions in vitro, proteins in the N phase exchange very slowly. The authors speculate that this is due to phosphorylation of the central IDR. However, they have not tested IDR phosphorylation in vitro. This could be done using an appropriate kinase such as SRPK or by mutation of specific serines to aspartate. I note that the SRPK structure was solved in the lab of Gourisankar Ghosh in the same department at UCSD and may be a source of an appropriate clone of the kinase.

Again with respect to intracellular and in vitro condensates, the authors also observed that condensates dissipated at higher concentrations of RNA. It is not clear whether the RNA concentration is important or the ratio of N to RNA. Could the intracellular environment be reflective of a high RNA environment? Will tRNA lead to disruption of the liquid crystal phase.

Finally, the cryo-tomography observations need further explanation and analysis. Are they granular or just not uniform? Will a Fourier transform of the complex lead to a powder pattern consistent with a granular aggregate?

Reviewer #2 (Remarks to the Author):

The manuscript by Lu et al. focuses on the phase separation behavior of SARS-CoV-2 nucleocapsid (N) protein in the presence of viral RNA and membrane (M) proteins using primarily in vitro approaches. They also include validating N protein phase separation in cells, which is an important component of the work. The studies represent a novel set of studies on a timely topic of high

interest. The authors did a series of studies using various domain combinations to identify which regions were involved in the various phase separation behaviors observed in the in vitro assays. The demonstration that the SR-IDR is critical to phase separation with RNA and that the dimerization domain and C-terminal IDR is critical for phase separation with M protein (independent of RNA) are important advances. Additionally, the exclusive condensate subdomains containing N protein-RNA and N protein-M protein is an interesting result although the physiological relevance of this result is somewhat unclear. Overall, there are novel and important findings in this work, however, there are some relatively moderate concerns that need to be adequately addressed.

The in-cell work is arguably one of the most important findings since it lends support to N protein condensate formation in cells. The finding that in-cell N protein phase separation is apparently quite different from the in vitro results is notable and highlights the potential differences between these approaches to studying phase separation. The authors refer the results from Carlson et al (a bioRxiv pre-print) and suggest that this could be due to differences in phosphorylation of the SD-IDR. However, this is highly speculative and should be directly assessed. Is the N protein in fact phosphorylated in the cell model system? The work would be further strengthened by demonstrating whether or not the mutually exclusive condensate structures seen with inclusion of M protein can be reproduced in cells. Without such data it remains unclear whether this is a biologically relevant finding or an artifact of the in vitro assay. This is important to clarify as the authors use the in vitro data to support their hypothetical model in Figure 6. Finally, the in-cell differences also beg the question of whether the domain differences in phase separation noted with N protein in vitro would be replicated in a more physiological setting.

The authors did a thorough assessment of different N protein domains in the phase separation with RNA and M protein, which is a strength of the paper. However, it would have been best to see an N protein that contains the SR-IDR and C-terminal domains (e.g. residues 175-419) to establish the impact of the SR-IDR on condensate formation with M-proteins.

Some experiments appear incomplete in Figure 2. For example, performing partial and full bleaching in FRAP experiments for all of the conditions would better delineate potential differences in the dynamic nature of the condensate structures. It is not clear why FRAP experiments were not performed on condensates in the N protein+RNA+M protein. It would be useful to know whether the mutually exclusive compartments are differentially dynamic.

The work would be strengthened by including some important controls. For example, GFP protein only controls (for the M protein-related experiments), unlabeled protein or RNAs (lacking either fluorescent molecule or GFP) and a control to determine whether the RNA effects are specific to viral RNA or not.

Experiments lack quantitative analyses (Figure 1A,C, 3, 5C-H, S1C,F, S2A, S3E, S4B) leaving much of the differences qualitative. Also, the authors mention "gel-like condensates" in a few places, but do not clearly indicate how they have determined the structures are in fact gel-like and not just displaying slower dynamic exchange of proteins.

Reviewer #3 (Remarks to the Author):

Lu et al. describe studies in which they attempt to demonstrate the propensities of SARS-CoV-2 N (N is used herein) to phase separate. The ability to phase separate in in vitro as well as in stable cell populations is demonstrated to be dependent on the central IDR domain in N, the expression of M, that forms a concentric outer layer on what appear to be condensates, but not on the presence of RNA (short 17-mer oligomers are generally used, but longer strands of viral RNA segments too).

The experiments and rationale are well laid out, with the authors citing several preprint (and non-peer reviewed) precedence to the work presented in the manuscript. This reviewer appreciates this effort. The number of preprint articles on the subject of N and phase separation abound and

slightly diminishes the novelty of the manuscript. The authors do not cite a FASEB J (A proposed role for the SARS-CoV-2 nucleocapsid protein in the formation and regulation of biomolecular condensates by Cascarina SM & Ross ED. FASEB J. 2020 Jun 20:10.1096/fj.202001351. doi: 10.1096/fj.202001351. PMID: 32562316), a published article describing bioinformatics analyses and the potential for N to mediate phase separation. This should be included.

The authors do not demonstrate many characteristics of LLPS (fission, fusion, mobility), while the FRAP experiments also demonstrate that the condensates are sluggish at recovering. How do we know these are phase separated condensates? Can these condensates be eliminated in a control experiment?

Further analyses of recovery after photobleaching need to be performed. The aspects of phase separation (fission, fusion, mobility) will be essential to demonstrate before further consideration of this article.

The expression studies are to be commended. A serious discussion on N over expression will be needed and citation of precedence to allay my concerns.

What is the stoichiometry of N:RNA (cellular RNA) in these cell lines expressing N?

Is RNA needed?

Are these stress granules or another type of N-containing granule?

Is the N-mediated condensation a result of overexpression and what is the threshold to see and not to see N-granules?

The phase contrast light microscopy images are of very poor quality and difficult to discern. These need to be reproduced.

Essentially, this paper is highly observational in nature. and raises the questions of "so what?" and what is the functional significance? The models for N-mediated encapsidation with RdRp, RNA and perhaps cellular partners remain standard fare in the current literature; nevertheless, this reviewer believes that the N-RNA-M condensate might very well be important in the replication cycle of SARS-CoV-2. Is there a way in which the authors could show some function for the N condensates: in polymerase function? in altering the RNA structure and readying it for packaging, for example?

An interesting and critical experiment would be to observe the fusion between the tri-condensate (N-RNA-M). Evidence for fission fusion and movement would be critical for any future consideration at Nat. Comm.

We thank the reviewers for their insightful comments and questions. In response to both the reviewer comments and our own further progress since the original submission, we have expanded the manuscript in three major areas. First, we have added several extra N protein variants to our in vitro analyses (presented in **Figures 3 and 5**), which further refine our assignment of functional regions within the N protein. Second, we generated both phospho-mimic and phosphorylation-resistant mutants of the N protein, and examined their behavior both in vitro and in cells. Third, we performed quantitative crosslinking mass spectrometry to identify particular regions within the N protein that mediate protein-protein (and likely protein-RNA) interactions within the observed condensates. These changes, plus others described below, make the manuscript a stronger, more complete work. Below we address each reviewer question in **blue text**. In the attached revised manuscript, substantive changes are highlighted in **purple text**.

Reviewer 1:

The authors show that N protein association and phase separation is dependent on both RNA and KCl concentration. RNA oligomers and long 5' UTR containing RNAs lead to spherical complexes. RNA with CoV packing signal lead to very large fibrillar structures. RNA oligomers can exchange from the N phase but N protein diffusion is restricted. The authors then show that the soluble domain of SARS CoV2 M protein also induces formation of a novel phase separation. The authors present striking micrographs of multi-layer particles with an external layer enriched in M and an internal layer enriched in RNA. With a series of truncation mutants the authors demonstrate that the phase separation activity can be attributed to the intrinsically disordered region (IDR) between the N-terminal and C-terminal folded domains. The authors propose a model of assembly where M helps organize N for binding to RNA and together they form the core of the nascent virus. Conversely, when expressed in mammalian cells, N protein does not form the sort of phase separation observed in vitro. They suggest that the intracellular N may be phosphorylated in the SR-rich segment of the IDP that is involved in association and may thus prevent the phase separation seen in the unphosphorylated protein in vitro.

Several groups have investigated the ability of SARS-COV2 N protein to undergo phase separation. The results demonstrated in this paper are timely and of interest. The experiments and results are intriguing. While I have many questions, I am enthusiastic about the manuscript.

1. While short oligos unsurprisingly diffuse in and out of the condensed phase, it seems unlikely that the larger RNAs can, but that should be ascertained. While short RNA oligos and longer RNAs from the 5'UTR yield round complexes, PS576, a 576 nucleotide RNA containing a putative packing site, yielded huge fibrillar structures. This effect is not investigated further. Will a larger RNA with the packaging site yield round complexes? Is the PS576 RNA more folded and compact? Are the PS576 complexes more less labile, or conversely can PS576 displace other RNA? It is not clear why the authors did not further investigate this RNA. Indeed, if an RNA with a packaging site yields a fibrillar structure, are the spherical particles artifactual? Will a combination of PS576 and M yield a fibrillar or compact particles.

We thank the reviewer for their overall enthusiasm for the manuscript, and for these carefully-considered thoughts. At the reviewer's behest, we have performed phase separation experiments with N protein and a 1000-nt viral RNA encompassing the packaging sequence (PS1000). This RNA forms fibrillar structures similar to that seen with the PS576 RNA, indicating that the RNA sequence does affect the overall architecture of the resulting condensates (see **Fig. 1** below). While we appreciate the reviewer's interest in further exploration of this phenomenon, we feel that a proper exploration of the effects of RNA sequence and RNA structure on condensate formation are outside the scope of this manuscript. Such an effort would require first establishing that this sequence does in fact contain the bona fide packing signal for SARS-CoV-2, followed by a detailed dissection of the exact region involved and a significant effort to determine the likely secondary structure of this

region. This is certainly important work, but must await a dedicated study. With respect to the reviewer's questions about whether the spherical structures formed with other RNAs are artifactual, we believe not. In agreement with other recent work (ref. 61) and as discussed in the Discussion, specific recognition of the packaging sequence (likely by the N-terminal RNA binding domain) is likely followed by "condensation" involving relatively non-specific RNA binding interactions from the IDRs and the C-terminal dimerization domain. In that sense, the spherical condensates are representative of the non-specific condensation step of viral RNA packaging. We did examine the morphology of N+M+PS576 condensates (Fig. S7c in the revised manuscript and Fig. 2 below). These show an intermediate morphology.

In cells, N forms highly dynamic condensates. While under selected experimental conditions *in vitro*, proteins in the N phase exchange very slowly. The authors speculate that this is due to phosphorylation of the central IDR. However, they have not tested IDR phosphorylation *in vitro*. This could be done using an appropriate kinase such as SRPK or by mutation of specific serines to aspartate. I note that the SRPK structure was solved in the lab of Gourisankar Ghosh in the same department at UCSD and may be a source of an appropriate clone of the kinase.

We thank the reviewer for this suggestion. Indeed, we were curious to follow up on this question as well, so we generated two mutant constructs for analysis: N^{145A} with all serine residues in the S/R rich region mutated to alanine, and N^{115D} with eleven of these serines mutated to aspartate to mimic the charge effects of phosphorylation. In the revised manuscript, we analyzed these mutants both in cells (Fig. 5 and Fig. S5) and *in vitro* (Fig. 3). *In vitro*, the phospho-mimetic N^{115D} mutant shows dramatically faster droplet fusion than wild-type or non-phosphorylatable N^{145A} protein, indicating that phosphorylation within the S/R rich region results in increased dynamics within condensates (Fig. 3r). In cells, the phospho-mimetic N^{115D} mutant shows similar dynamics as wild-type protein, while the non-phosphorylatable N^{145A} mutant shows slower dynamics (Fig. 5 and Fig. S5). These data strongly support the idea that N is highly phosphorylated in the S/R rich region in cells, and that phosphorylation in this region governs the viscosity of N+RNA condensates.

Again with respect to intracellular and *in vitro* condensates, the authors also observed that condensates dissipated at higher concentrations of RNA. It is not clear whether the RNA concentration is important or the ratio of N to RNA. Could the intracellular environment be reflective of a high RNA environment? Will tRNA lead to disruption of the liquid crystal phase.

The dissipation of condensates we observe at high RNA concentrations is typical of two-component mixtures of multivalent molecules forming condensates. As noted in our manuscript, this is termed

“re-entrant” behavior, and arises when one component (in this case, RNA) is present at much higher stoichiometry than the other. The extreme excess of binding sites in the predominant component suppresses formation of the two-component condensate. This phenomenon has been characterized in a number of biomolecular condensates and indeed was observed and discussed in detail in a related pre-print manuscript on SARS-CoV-2 N (ref. 61).

With respect to the point about RNA concentration in cells and how this could affect the phase separation behavior of the N protein, we can only speculate. RNA concentration in mammalian cells is quite high (very roughly, ~8,000 ng/uL), but the vast majority of this RNA is ribosomal (rRNA) and tRNA. At the same time, there is also a high concentration of RNA-binding proteins in the cell. We observe formation of phase separated condensates in cells, so the concentration of RNA available for N protein binding must be within a range conducive to this phase separation. With respect to the reviewer’s question about tRNAs, we don’t know whether the small, highly structured tRNAs would influence N protein condensation or not.

Finally, the cryo-tomography observations need further explanation and analysis. Are they granular or just not uniform? Will a Fourier transform of the complex lead to a powder pattern consistent with a granular aggregate?

Thanks for the suggestion. We performed a power spectrum analysis of both tomographic slices (**Fig. S1f**) and individual tilt images (**Fig. S1g**), but did not detect any features suggestive of strong periodicity. This analysis is not foolproof, however, since it only detects very strong and highly regular features. Thus, the question of whether there is regular internal structure in the N+RNA condensates remains one for future in-depth analysis.

Reviewer 2:

The manuscript by Lu et al. focuses on the phase separation behavior of SARS-CoV-2 nucleocapsid (N) protein in the presence of viral RNA and membrane (M) proteins using primarily in vitro approaches. They also include validating N protein phase separation in cells, which is an important component of the work. The studies represent a novel set of studies on a timely topic of high interest. The authors did a series of studies using various domain combinations to identify which regions were involved in the various phase separation behaviors observed in the in vitro assays. The demonstration that the SR-IDR is critical to phase separation with RNA and that the dimerization domain and C-terminal IDR is critical for phase separation with M protein (independent of RNA) are important advances. Additionally, the exclusive condensate subdomains containing N protein-RNA and N protein-M protein is an interesting result although the physiological relevance of this result is somewhat unclear. Overall, there are novel and important findings in this work, however, there are some relatively moderate concerns that need to be adequately addressed.

The in-cell work is arguably one of the most important findings since it lends support to N protein condensate formation in cells. The finding that in-cell N protein phase separation is apparently quite different from the in vitro results is notable and highlights the potential differences between these approaches to studying phase separation. The authors refer the results from Carlson et al (a bioRxiv pre-print) and suggest that this could be due to differences in phosphorylation of the SD-IDR. However, this is highly speculative and should be directly assessed. Is the N protein in fact phosphorylated in the cell model system? The work would be further strengthened by demonstrating whether or not the mutually exclusive condensate structures seen with inclusion of M protein can be reproduced in cells. Without such data it remains unclear whether this is a biologically relevant finding or an artifact of the in vitro assay. This is important to clarify as the authors use the in vitro data to support their hypothetical model

in Figure 6. Finally, the in-cell differences also beg the question of whether the domain differences in phase separation noted with N protein in vitro would be replicated in a more physiological setting.

We thank the reviewer for their insights. Based on prior studies with related betacoronaviruses (referenced in the manuscript), the N protein is hyperphosphorylated in cells on serine residues in the SR region and the C-terminal domain. A new pre-print manuscript on *bioRxiv* from Ahmet Yildiz's group at UC Berkeley (ref. 54) shows that SARS-CoV-2 N protein purified from mammalian cells is highly phosphorylated. While this group could map a number of phosphorylation sites on the N protein by mass spectrometry, they were unable to map the SR region due to sequence features: specifically, the peptides resulting from trypsin digestion were too short to analyze by mass spectrometry. Nonetheless, as we note above in response to a question from Reviewer #1, our new data on the in vitro (**Fig. 3**) and in-cell (**Fig. 5**) behavior of phospho-mimic N^{115D} and non-phosphorylatable N^{145A} mutants strongly supports the idea of phosphorylation in the S/R rich region regulating the viscosity of N+RNA condensates.

We thank the reviewer for their suggestion to test phase separation of M protein with N protein in cells. However, we feel that such an analysis is outside the scope of this work. A proper analysis of this question would entail the construction of cell lines expressing labeled N protein, labeled M protein (being membrane-localized, identifying the proper construct for this labeling may not be straightforward), and ideally a large fragment of the SARS-CoV-2 genome (also tagged, perhaps with MS2 hairpins or similar). We are excited to pursue this strategy in future work.

Given the agreement between in vitro and in vivo behavior of our phospho-mimic and phosphorylation-resistant N protein variants, we feel confident that our in vitro domain analysis in **Fig. 3** and **Fig. 6** is biologically relevant. Our primary goal with these studies was to study how the N protein behaves in the context of viral RNA packaging; prior studies have shown that betacoronavirus N proteins are hypophosphorylated in this context.

The authors did a thorough assessment of different N protein domains in the phase separation with RNA and M protein, which is a strength of the paper. However, it would have been best to see an N protein that contains the SR-IDR and C-terminal domains (e.g. residues 175-419) to establish the impact of the SR-IDR on condensate formation with M-proteins.

Thanks for the suggestion. We have generated two new truncations (N¹⁷⁵⁻⁴¹⁹ and N¹⁷⁵⁻³⁶⁴) and found that both could form condensates with RNA (**Fig. 3**) and the M protein (**Fig. 6**).

Some experiments appear incomplete in Figure 2. For example, performing partial and full bleaching in FRAP experiments for all of the conditions would better delineate potential differences in the dynamic nature of the condensate structures. It is not clear why FRAP experiments were not performed on condensates in the N protein+RNA+M protein. It would be useful to know whether the mutually exclusive compartments are differentially dynamic.

We apologize for the confusion on this point. For N protein + 17-mer RNA condensates, full bleaching experiments are presented in **Fig. 2a-b**, and partial bleach experiments are presented in **Fig. S2a-b**. For N protein + viral RNA fragments, both full and partial bleaching experiments are presented in **Fig. 2c-d**.

We also performed FRAP on the N+M+RNA condensates, and found that all three components show slow dynamics (**Fig. S7e**).

The work would be strengthened by including some important controls. For example, GFP protein only controls (for the M protein-related experiments), unlabeled protein or RNAs (lacking either fluorescent molecule or GFP) and a control to determine whether the RNA effects are specific to viral RNA or not.

The reviewer makes an important point about including a GFP-only control in our experiments with GFP-M¹⁰⁴⁻²²². We have added this control to **Fig. S6d** in the revised manuscript. As expected, GFP alone does not induce condensation of the N protein.

For experiments with N protein and RNA, the majority of our in vitro condensation assays have in fact been done with unlabeled N protein, which shows no difference in behavior compared to labeled protein in our hands. We would point out that the label in this case is a fluorescent dye linked to an introduced cysteine residue at the protein's N-terminus, which is a very minor perturbation compared to, say, fusion to a fluorescent protein. Similarly, a single fluorophore attached to the end of an RNA 265-1000 nucleotides in length is a minimal perturbation and is not expected to alter behavior. Many other groups have also performed phase separation analysis with unlabeled N protein and unlabeled RNA (Cubuk, et al., bioRxiv; Carlson, et al., bioRxiv, 2020; Perdikari, et al., bioRxiv, 2020).

With respect to RNA sequence, we have established that the N protein can form condensates with short RNAs (17-mer) and longer viral RNA derived fragments. Other groups have demonstrated that N protein condensates can be induced by diverse RNA, including poly-U (Cubuk, et al., bioRxiv, 2020; Savastano, et al., bioRxiv, 2020), poly-C (Jack, et al., bioRxiv, 2020), poly-A (Jack, et al., bioRxiv, 2020), and other non-specific RNAs (Carlson, et al., bioRxiv, 2020; Perdikari, et al., bioRxiv, 2020; Iserman, et al., bioRxiv, 2020). We know there is much to be done to understand how the specificity is achieved for viral RNA in cells, which is our future direction and goal.

Experiments lack quantitative analyses (Figure 1A,C, 3, 5C-H, S1C,F, S2A, S3E, S4B) leaving much of the differences qualitative. Also, the authors mention “gel-like condensates” in a few places, but do not clearly indicate how they have determined the structures are in fact gel-like and not just displaying slower dynamic exchange of proteins.

We thank the reviewer for these insights. We developed a quantitation metric based on the total amount of protein or RNA fluorescence per microscopic field that is found within condensates versus outside condensates. We used this method to quantify the phase separation capacity of full-length and truncated N with RNA (**Fig. 1a** and **Fig. 3**) and with M protein (**Fig. 6**). These analyses have enabled us to draw more subtle distinctions between various mutants, and we appreciate the prodding from the reviewer to perform this more in-depth analysis.

We agree with the reviewer that our terminology was potentially misleading in the original submission. We have therefore minimized the use of phrases “liquid-like”, “gel-like”, and “solid-like” throughout the manuscript, instead discussing condensate behavior in terms of higher or lower viscosity.

Reviewer #3 (Remarks to the Author):

Lu et al. describe studies in which they attempt to demonstrate the propensities of SARS-CoV-2 N (N is used herein) to phase separate. The ability to phase separate in in vitro as well as in stable cell populations is demonstrated to be dependent on the central IDR domain in N, the expression of M, that forms a concentric outer layer on what appear to be condensates, but not on the presence of RNA (short 17-mer oligomers are generally used, but longer strands of viral RNA segments too).

The experiments and rationale are well laid out, with the authors citing several preprint (and non-peer reviewed) precedence to the work presented in the manuscript. This reviewer appreciates this effort. The number of preprint articles on the subject of N and phase separation abound and slightly diminishes the novelty of the manuscript. The authors do not cite a FASEB J (A proposed role for the SARS-CoV-2 nucleocapsid protein in the formation and regulation of biomolecular condensates by Cascarina SM & Ross ED. FASEB J. 2020 Jun 20;10.1096/fj.202001351. doi: 10.1096/fj.202001351. PMID: 32562316), a published article describing bioinformatics analyses and the potential for N to mediate phase separation. This should be included.

We apologize for omitting this important contribution. We added this reference in our updated manuscript (ref. 42).

The authors do not demonstrate many characteristics of LLPS (fission, fusion, mobility), while the FRAP experiments also demonstrate that the condensates are sluggish at recovering. How do we know these are phase separated condensates? Can these condensates be eliminated in a control experiment?

We feel confident that the evidence presented in our original submission strongly supported the conclusion that these are in fact phase separated condensates. For example, we showed droplet fusion (albeit with slow dynamics; **Fig. 1d**), free diffusion of at least small RNAs in the droplets by FRAP analysis (**Fig. 2a**), and diffusion of new N protein into existing condensates (**Fig. S2c**).

In the revised manuscript, this case is made even more clear. With our N^{115D} and N^{145A} mutants, we now show that the liquidity of the N+RNA condensates is directly related to the phosphorylation state of the protein. For example, droplets of N^{115D} fuse with much faster dynamics than those of wild-type N or N^{145A} in vitro (**Fig. 3r**), and our FRAP analysis in cells further supports this conclusion (**Fig. 5, S5**). We also now show that N+RNA droplets are dissolved within seconds of adding 1,6-hexanediol, an aliphatic alcohol reported to be able to dissolve liquid- and gel-like phase separated structures (Posey, et al., *Methods Enzymol.* **611**, 1–30 (2018) (**Fig. 1e**).

Further analyses of recovery after photobleaching need to be performed. The aspects of phase separation (fission, fusion, mobility) will be essential to demonstrate before further consideration of this article.

In the revised manuscript, we have added further FRAP analysis, both in vitro and in cells. We refer the reviewer to the above response regarding the various aspects of phase separation that we have clearly demonstrated.

The expression studies are to be commended. A serious discussion on N over expression will be needed and citation of precedence to allay my concerns. What is the stoichiometry of N:RNA (cellular RNA) in these cell lines expressing N? Is RNA needed? Are these stress granules or another type of N-containing granule? Is the N-mediated condensation a result of overexpression and what is the threshold to see and not to see N-granules?

As noted in the response to Reviewer #1 above, the RNA concentration in cells is much higher than that used for our in vitro analysis. The vast majority of this RNA is ribosomal RNA and tRNA, however, and the remaining RNA is likely bound by many other RNA binding proteins in the cell. Therefore, it is near-impossible to quantitatively compare the behavior of the N protein in cells with our detailed in vitro analyses.

In our cellular experiments, we did not perform any perturbations to induce the formation of stress granules. As noted in a published study that we cite in the manuscript, the SARS-CoV-2 N protein associates with stress granule proteins including G3BP1 and G3BP2 (Gordon, et al., Nature, 2020). These and other data suggest that the N protein, when overexpressed in cells outside the context of an active viral infection, associates with cellular RNA (and stress granules) in a relatively non-specific way. Finally, as with any macromolecule undergoing phase separation, the level of N in cells is a critical factor in its capacity to form condensates. This can be observed qualitatively in our analyses of U2OS cells expressing Clover-tagged N, where a minority of cells show condensates, and these cells tend to show higher overall levels of expression than cells without condensates (Fig. 5a). Unfortunately the system we used for overexpression does not allow for the fine-grained expression control that would be needed to adequately explore how N protein concentration relates to phase separation in cells.

The phase contrast light microscopy images are of very poor quality and difficult to discern. These need to be reproduced.

We respectfully disagree with the reviewer on this point. When compared to DIC images in previously published papers of phase separated droplets in vitro (Fig. 3, left below), our images are of similar quality (Fig. 4, right below). We would note here that many of our images show subtle distortions arising from droplets that are not in the plane of focus; DIC imaging is particularly susceptible to this effect, while fluorescence imaging is less sensitive. In many publications using DIC imaging for characterization of phase separated droplets, imaging is only performed after all droplets have settled to the microscope slide, and are therefore all on the same plane. Since one of our main goals was to compare the behavior of many N protein variants (for example in Figs. 3 and 6), we opted to perform imaging after a set amount of incubation time, rather than waiting for full settling of droplets. Finally, even in the figure panels most affected by out-of-focus droplets, the image quality is more than sufficient to observe droplets correlated with the fluorescence images, and all quantitation is done with the fluorescence images.

Fig. 3. Published DIC images of in vitro phase separated condensates from four independent groups.

Fig. 4. DIC images and fluorescence images of N+RNA droplets in our study.

Essentially, this paper is highly observational in nature. and raises the questions of "so what?" and what is the functional significance? The models for N-mediated encapsidation with RdRp, RNA and perhaps cellular partners remain standard fare in the current literature; nevertheless, this reviewer believes that the N-RNA-M condensate might very well be important in the replication cycle of SARS-CoV-2. Is there a way in which the authors could show some function for the N condensates: in polymerase function? in altering the RNA structure and readying it for packaging, for example?

These are all very important questions. With regard to directly showing function in viral pathogenesis, this is unfortunately outside the scope of the study. As referenced in our manuscript, the N protein has been shown to contribute to RNA polymerase function in related coronaviruses. While it remains unknown whether the protein's ability to form condensates with RNA is

functionally relevant in this context, we think it's likely. Similarly, our observation that the N protein forms condensates with the M protein is likely to be functionally important for viral packaging. However, we are currently unable to perform the experiments that would be needed to test this idea, such as engineering live virus to determine if N protein C-terminal truncations compromise virion biogenesis. Nonetheless, our in vitro reconstitution of N+RNA, N+M and N+M+RNA condensate formation both provides important new biological insights, and points toward in vitro assays that could be used to target these steps using small molecules or other interventions.

An interesting and critical experiment would be to observe the fusion between the tri-condensate (N-RNA-M). Evidence for fission fusion and movement would be critical for any future consideration at Nat. Comm.

We did observe fusion between the N+M+RNA condensates, with very slow dynamics (Fig. 5 below). There is a tendency that the M protein is expelled to the surface of the three-component condensates, as might be expected from our observations that RNA and M mutually repel one another in the condensates. While this analysis is not included in our revised manuscript, we can certainly add it if the reviewer and editor feel strongly that it should be included.

REVIEWER COMMENTS

Reviewer #1 (Remarks to the Author):

SarsCoV2 N protein has two ordered domains flanked by intrinsically disordered regions, IDRs. In the original manuscript, the authors identified the SR region of the central IDR as critical to nucleic acid and M protein-dependent condensate formation. The original draft was exciting but incomplete. In the revised draft, the authors have incorporated investigations of the importance of IDR phosphorylation on the regulation of condensate formation in vitro, and correlated intracellular behavior with in vitro phosphorylation. The paper is a much more complete story and will be a highly significant contribution to the literature.

In my initial review, I focused on omission regarding the role of the RNA substrate on condensate morphology and on the disparity between condensate viscosity in vitro and in vivo. The authors did expand their substrate selection in vitro and make a reasonable argument that a more complete analysis of RNA substrate is a story on its own and beyond the scope of this publication.

The authors provided a much more complete analysis of the difference in condensate viscosity. In the first manuscript they suggested the difference would relate to phosphorylation state of the SR domain. They developed a non-phosphorylatable mutant with 14 S \rightarrow A mutation and a constitutive mimic of the phosphorylated state with 11 S \rightarrow D mutations. In cells, the S \rightarrow A mutant behaves shows high viscosity, as seen with unphosphorylated N in vitro. Conversely, the S \rightarrow D mutant shows low viscosity behavior in vitro and in cell like non-mutant N and in vitro. This firmly supports the authors' hypothesis that viscosity is regulatable and provides a unique tool for future investigators to regulate N behavior.

In addition, The authors include a crosslinking study that demonstrates the basis of N-N interaction, showing that crosslinking occurs more rapidly in the presence of RNA than in its absence.

I am satisfied with the authors' revisions.

There are some minor issues the authors should address:

In Figure 4, the program used for calculating the electrostatic surface is not indicated. It could be a rigorous calculation or a Coulombic surface in vacuo. The authors should add sufficient detail.

References should be added regarding the generalized behavior of hexanediol. The text around line 130 should note that disruption was observed in 10% hexanediol.

Line 182, the change in fusion rate of N may be function of viscosity, but at a molecular level may be affected by protein RNA interaction, which should also be considered.

Line 193 BS3 needs to be better described in text or methods, including its full chemical name and source. The deuteration needs to be explained in results or methods without expecting the reader to understand ,

Reviewer #2 (Remarks to the Author):

The authors have addressed some of the initial critical concerns and provides some additional clarifications that were helpful. The new data provided helps build a stronger manuscript. However, there is a general issue with the level of rigor in some of the data reported. The following are seen as remaining issues that need to be addressed:

- There are no error bars in the data presented in Figure S1b and S1c. Why are error bars not presented? How many times, if at all, were these experiments repeated.
- There is no statistical comparisons of data in Figure 2b-d for differences between the RNA dynamics or differences between N protein and RNA dynamics. This interferes with the ability of the author's to make statements such as "but in these condensates the longer RNAs showed much slower dynamics than the N protein, with the longest viral RNA showing the lowest mobility".
- The paper still lacks FRAP analysis of PS576 RNA condensates.
- The differences in the rate of fusion between N14SA and N11SD proteins is not rigorously assessed and requires some quantitation (Figure 3r).
- The authors should better characterize the structures containing the N protein in the cell studies. Are structures stress granules, are stress granule proteins present and/or is RNA present in the structures.
- It is difficult to judge the rigorousness of the in vitro phase separation quantitation experiments. How many times were these experiments repeated, how consistent were the results, etc.

Response to Reviewers

We thank the reviewers for their careful analysis and consideration of our revised manuscript. In our newly revised manuscript, we have added additional data in three areas. First, at the behest of Reviewer #2, we have added in vitro FRAP analysis of N+PS576 condensates (**Figure 2e**). Second, we have added additional analysis in cells treated with chemical inhibitors of two kinases, SRPK and GSK3, which support our conclusions about the role of N phosphorylation in regulating condensate dynamics (**Figure S6**). Finally, we have performed additional analyses to determine how the cellular N protein condensates are related to stress granules. These data, presented in the new **Figure 6**, reveal that SARS-CoV-2 N condensates recruit the stress granule core protein G3BP1, but not other known stress granule proteins. This behavior parallels the known ability of several viruses to sequester G3BP1 in order to inhibit stress granule formation, and identifies an exciting new function for SARS-CoV-2 N.

Below we address each remaining reviewer question in **blue text**. In the attached revised manuscript, substantive changes from the prior revised version are highlighted in **purple text**.

Reviewer #1

SarsCoV2 N protein has two ordered domains flanked by intrinsically disordered regions, IDRs. In the original manuscript, the authors identified the SR region of the central IDR as critical to nucleic acid and M protein-dependent condensate formation. The original draft was exciting but incomplete. In the revised draft, the authors have incorporated investigations of the importance of IDR phosphorylation on the regulation of condensate formation in vitro, and correlated intracellular behavior with in vitro phosphorylation. The paper is a much more complete story and will be a highly significant contribution to the literature.

In my initial review, I focused on omission regarding the role of the RNA substrate on condensate morphology and on the disparity between condensate viscosity in vitro and in vivo. The authors did expand their substrate selection in vitro and make a reasonable argument that a more complete analysis of RNA substrate is a story on its own and beyond the scope of this publication.

The authors provided a much more complete analysis of the difference in condensate viscosity. In the first manuscript they suggested the difference would relate to phosphorylation state of the SR domain. They developed a non-phosphorylatable mutant with 14 S-A mutation and a constitutive mimic of the phosphorylated state with 11 S-D mutations. In cells, the S-A mutant behaves shows high viscosity, as seen with unphosphorylated N in vitro. Conversely, the S-D mutant shows low viscosity behavior in vitro and in cell like non-mutant N and in vitro. This firmly supports the authors' hypothesis that viscosity is regulatable and provides a unique tool for future investigators to regulate N behavior.

In addition, The authors include a crosslinking study that demonstrates the basis of N-N interaction, showing that crosslinking occurs more rapidly in the presence of RNA than in its absence.

I am satisfied with the authors' revisions.

We thank the reviewer for their suggestions and criticisms of our original manuscript, which prompted us to significantly strengthen the revised manuscript as they note.

There are some minor issues the authors should address:

In Figure 4, the program used for calculating the electrostatic surface is not indicated. It could be a rigorous calculation or a Coulombic surface in vacuo. The authors should add sufficient detail.

We apologize for the omission. We calculated this surface using the APBS (Adaptive Poisson-Boltzmann Solver) plugin in PyMOL, which rigorously calculates the electrostatic surface using a continuum solvation model. This is now noted in the legend to **Figure 4**.

References should be added regarding the generalized behavior of hexanediol. The text around line 130 should note that disruption was observed in 10% hexanediol.

We have added the additional detail requested, and added references regarding the behavior of hexanediol towards biomolecular condensates.

Line 182, the change in fusion rate of N may be function of viscosity, but at a molecular level may be affected by protein RNA interaction, which should also be considered.

We agree with the reviewer on this point. We have added a sentence to this paragraph noting that the differences in viscosity may arise from modulation of either protein-RNA or protein-protein interactions.

Line 193 BS3 needs to be better described in text or methods, including its full chemical name and source. The deuteration needs to be explained in results or methods without expecting the reader to understand.

We have added the full chemical name of BS3 (bis(sulfosuccinimidyl)suberate), and provided a more approachable explanation of the molecular weight difference between BS3-d0 and BS3-d4, and how this difference enables quantitative comparisons in mass spectrometry experiments.

Reviewer #2

The authors have addressed some of the initial critical concerns and provides some additional clarifications that were helpful. The new data provided helps build a stronger manuscript. However, there is a general issue with the level of rigor in some of the data reported. The following are seen as remaining issues that need to be addressed:

There are no error bars in the data presented in Figure S1b and S1c. Why are error bars not presented? How many times, if at all, were these experiments repeated.

We regret having included a preliminary figure instead of the final experiment, which was performed in triplicate. We have replaced **Figure S1b-c** with these final data, which broadly show the same results as our original panels.

There is no statistical comparisons of data in Figure 2b-d for differences between the RNA dynamics or differences between N protein and RNA dynamics. This interferes with the ability of the author's to make statements such as "but in these condensates the longer RNAs showed much slower dynamics than the N protein, with the longest viral RNA showing the lowest mobility".

We now report percent recovery for both N protein and RNA for these experiments, and we also report these numbers in the text. Due to the extremely slow recovery rates for the N protein and the long RNAs, calculation of recovery half-time is not possible for these experiments.

The paper still lacks FRAP analysis of PS576 RNA condensates.

We have now added this analysis (**Figure 2e**). The results show that despite the morphological difference between N+RNA droplets with PS576 versus UTR265/UTR1000, the dynamics of the droplets are similar for all three RNAs.

The differences in the rate of fusion between N14SA and N11SD proteins is not rigorously assessed and requires some quantitation (Figure 3r).

This is a difficult question, since droplet fusion is rare and we have imaged only a few events for each protein from which to draw conclusions. That said, we examined three independent fusion events for both N11SD and N14SA, and observe 10-40 second fusion time (measured from initial contact to fully-round final droplet) for N11SD, compared to >30 minutes for N14SA (these are difficult to quantify exactly due to the time required for full fusion). Thus, the two proteins differ by at least 60-fold in their fusion rate. We have noted these numbers in the main text and legend to **Figure 3r**.

The authors should better characterize the structures containing the N protein in the cell studies. Are structures stress granules, are stress granule proteins present and/or is RNA present in the structures.

We agree with the reviewer that this is an important question. We performed an initial analysis by co-staining for known stress granule proteins (revised **Figure 6**). We find that N protein condensates contain RNA and also contain the core stress granule protein G3BP1. This finding is supported by several recent mass spectrometry studies that identify G3BP1 and G3BP2 as N protein interactors. Curiously, N protein condensates largely do not contain other known stress granule proteins like UBAP2L, the RNA helicase DDX1, or the ribosomal factor EIF3 η (EIF3e). Thus, it appears that instead of partitioning into stress granules, the N protein forms de novo condensates that sequester G3BP1. This behavior parallels that of many other viruses, which inhibit stress granule formation by sequestering G3BP1 (references in the revised manuscript). This conclusion is also supported by a recently-posted preprint manuscript on *bioRxiv* from the Greenblatt group, which shows that SARS-CoV-2 N binds G3BP1 and G3BP2, inhibits formation of stress granules, and alters host gene expression by binding host mRNAs (<https://doi.org/10.1101/2020.10.23.342113>).

This is obviously a very interesting and exciting finding, which reveals a potential third role for SARS-CoV-2 N in stress granule inhibition, in addition to its known roles in viral RNA production and packaging. A full analysis of this finding, however, is outside the scope of the current manuscript and must await future studies.

It is difficult to judge the rigorosity of the in vitro phase separation quantitation experiments. How many times were these experiments repeated, how consistent were the results, etc.

Our in vitro phase separation experiments are highly reproducible, with consistent results across protein preps, RNA batches, and microscope sessions. Most experiments were performed multiple times in order to optimize imaging conditions, with similar results. Samples presented in a given figure panel (e.g. **Figure 1a**) were prepared and imaged together to maximize consistency.

REVIEWERS' COMMENTS

Reviewer #2 (Remarks to the Author):

The authors have performed additional experiments, added requested data and made necessary clarifications. In my opinion, the previous issues have been adequately addressed. Only a few very minor comments remain. The summary paragraph of the introduction does not highlight the in-cell findings well. The sentence in lines 113-115 is not complete. Overall, this manuscript is substantively improved